# MAESTRO: MASKED ENCODING SET TRANSFORMER WITH SELF-DISTILLATION

**Matthew E. Lee**[1,2,3], **Jaesik Kim**[1,2], **Matei Ionita**[1], **Jonghyun Lee**[1,2], **Michelle McKeague**[1],
**Yonghyun Nam**[1,2], **Irene Khavin**[1], **Yidi Huang**[1,2], **Victoria Fang**[1,3], **Sokratis Apostolidis**[1],
**Divij Mathew**[1,3], **Shwetank**[1], **Ajinkya Pattekar**[1], **Zahabia Rangwala**[1], **Amit Bar-Or**[1],
**Benjamin A. Fensterheim**[1], **Benjamin Abramoff**[1], **Rennie Rhee**[1], **Damian Maseda**[1],
**Allison R. Greenplate**[1,*] **E. John Wherry**[1,3,*] **Dokyoon Kim**[1,2,*]

[1]Institute for Immune Health and Immunology
[2]Department of Biostatistics Epidemiology and Informatics
[3]Department of Systems Pharmacology & Translational Therapeutics
University of Pennsylvania
Philadelphia, PA 19104, USA
{matthew.lee1}@pennmedicine.upenn.edu

## ABSTRACT

The immune system is a complex network of cells, orchestrating coordinated responses throughout the human body over a person's lifetime. Cytometry enables profiling of this network, but current approaches focus on enumerating and phenotyping immune cells, failing to quantify the system as a whole. We present MAESTRO, a self-supervised set representation learning model that generates vector representations of set-structured data, which we apply to learn immune profile representations. Unlike previous studies that learn cell-level representations, MAESTRO uses all of a sample's cells to learn a set representation. MAESTRO leverages attention mechanisms to handle sets of variable number of cells and ensure permutation invariance, coupled with a self-distillation framework. It is capable of reconstructing immune profiles (cells) even when 90% are hidden as its training objective. We benchmark our model against existing cytometry approaches and other existing machine learning methods that have never been applied in cytometry. Our model outperforms existing approaches in retrieving cell-type distributions and capturing clinically relevant features for downstream tasks such as disease diagnosis, age, sex. Code available: https://github.com/matthew-lee1/MAESTRO

## 1 INTRODUCTION

Cytometry, a single-cell technology, has revolutionized biomedical research by enabling high-dimensional profiling of individual cells within biological systems (Bendall et al., 2011). Cytometry provides information about cell-types, states, and functions, which is crucial for understanding immune responses, disease mechanisms, and therapeutic interventions (Spitzer & Nolan, 2016). One of the primary challenges in analyzing cytometry data is the number of cells and features of the data (Wang et al., 2023). Hundreds of thousands of cells are collected for each sample, where each cell is characterized by measurements across tens of proteins, resulting in complex datasets that are difficult to process and interpret using conventional methods. Processing large-scale single-cell datasets demands substantial resources and efficient algorithms capable of scaling to millions of cells without compromising performance (Kimball et al., 2018). Variable set sizes—due to differences in the number of cells collected from each sample—complicate the application of traditional deep learning models that expect fixed-size inputs. Previous methods mitigate this issue by randomly sampling cells to a fixed number, limiting the use of all available data (Hu et al.). Furthermore, cytometry data exhibits permutation invariance, meaning the ordering of cells within a sample is arbitrary and carries no intrinsic information. Models must be robust to such permutations to avoid

---

*These authors contributed equally to this work.

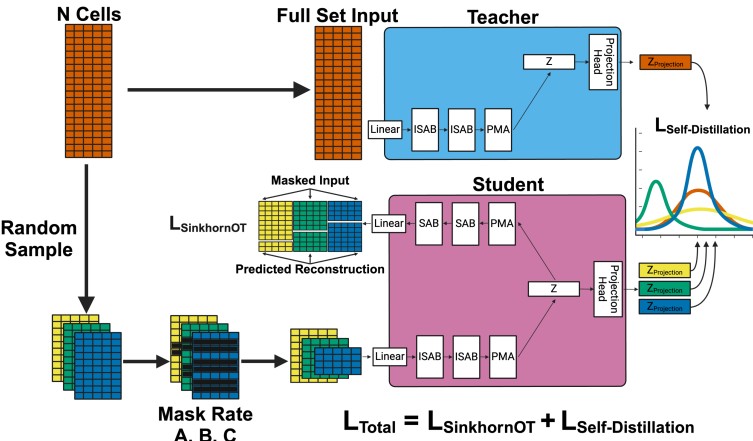

Figure 1: **MAESTRO Architecture.** N cells are sampled due to computational constraints. The sampled subset is fed into a student model, while the full set is fed to a teacher model. The teacher model only requires an encoder to provide a stable target of the full input set for the student model. The teacher encoder parameters are the exponential moving average of the student's, therefore it does not have the same computational constraints. For the student model, we apply masking at various rates to promote diverse learning and use a Set Transformer backbone for encoding, followed by a permutation-invariant decoding and reconstruction mechanism. We use non-linear projection heads on the latent embeddings to align representations of the full set from the teacher and the masked subset from the student. ISAB, PMA, and SAB blocks are defined in 3.1.

spurious correlations. Existing methods may fail under these demands, highlighting the need for more scalable solutions.

Self-supervised learning (SSL) is a powerful paradigm for learning representations of large, high-dimensional datasets (Chen et al., 2020; Radford et al., 2021; Kramer, 1991). In single-cell analysis, SSL methods have demonstrated success in capturing cellular states, trajectories, and functional characteristics, outperforming traditional gating methods, dimensionality reduction techniques, supervised approaches, and standard statistical modeling (Cui et al., 2024; De Donno et al., 2023; Bunne et al., 2024). Although current SSL models excel at understanding individual cells, there is a significant gap in their application to cell populations, as previous methods were designed for individual cells without incorporating sample membership information, resulting in the loss of cell population-level insights. This limitation hinders the extraction of meaningful, compact, and computationally practical representations of the entire cellular milieu in samples, making the discovery of multicellular interactions difficult. Consequently, set representation learning approaches are needed to effectively encode and represent cell populations.

Current set representation learning methods are inadequate for cytometry data because they either struggle to manage the number of cells inherent to such datasets, or rely on supervised learning approaches instead of self-supervised methods requiring extensive labels (Van Gassen et al., 2016). Specifically, some approaches maintain permutation invariance but struggle with the size of cytometry data, limiting their effectiveness (Lee et al., 2019; Mialon et al., 2022). Conversely, other methods can handle the size of cytometry, but do not preserve permutation invariance, making their representations sensitive to the ordering of cells within data matrices (Lecun et al., 2015). Additionally, many existing techniques are supervised, requiring labeled data, which is often scarce in single-cell studies and restricts their ability to leverage large, unlabeled datasets effectively (Hu et al.). These combined limitations prevent current set representation learning methods from effectively processing and representing cytometry data in single-cell analyses.

We propose MAESTRO (Masked Encoding Set TRansformer with Self-DistillatiOn), a self-supervised architecture for generating vector representations of set-structured data (see Figure1). MAESTRO employs attention mechanisms tailored for sets, including multi-head attention pooling (Lee et al., 2019), to handle large, variable-sized datasets while ensuring permutation invariance. By integrating a masked encoding strategy and a self-distillation framework, it learns informative rep-

resentations without extensive labels. We validate MAESTRO through experiments on both global (set-level) and local (element-level) representations, demonstrating its ability to encode sample-wide features and individual elements that distinguish phenotypes (Chan et al., 2021). Using linear probing, we predict sample-level features like disease phenotypes from the embeddings and assess cell-level representations by retrieving cell-type distributions without labels. This showcases MAESTRO's proficiency in encoding fine-grained, high-dimensional cellular information relevant to immune profiles.

To the best of our knowledge, MAESTRO is the first attention-based self-supervised set representation learning architecture capable of handling sets on the order of hundreds of thousands of elements and variable-sized sets in the context of single-cell data. Our key contributions are as follows:

- We introduce MAESTRO, a self-supervised set representation learning architecture that effectively generates vector representations of immune profiles from cytometry data.

- We developed MAESTRO that addresses the size of biological datasets that were previously too large to handle. Our method works with sets containing hundreds of thousands of elements, expanding what's possible in machine learning and data analysis.

- We evaluate MAESTRO against the state-of-the-art cytometry analysis method, manual gating, as well as against novel approaches like Deep Sets, Set Transformer, and OTKE (mentioned below) that have not previously been applied to cytometry data.

## 2 RELATED WORK

Current methods for analyzing cytometry data as a set include manual gating, DeepCyTOF, and clustering algorithms to identify cell populations (Mathew et al., 2020; Hu et al.; Van Gassen et al., 2015). Manual gating is a method where experts visually identify and select cell populations of interest on scatter plots or histograms based on their characteristic protein expressions. Manual gating is labor intensive and subject to operator bias due to manual thresholding of expression levels. Deep-CyTOF uses a 1-dimensional convolutional neural network with pooling, which, due to architecture design is effectively equivalent to Deep Sets (Zaheer et al., 2017). Automated clustering methods offer scalability, but may not capture complex cellular heterogeneity (Van Gassen et al., 2015; Traag et al., 2019) and are sensitive to parameter settings, such as the number of clusters (Pierson & Yau, 2015). For immunology research, manual gating is the state-of-the-art for analyzing cytometry data at the cellular level as demonstratedy by ground truth labels in (Kim et al., 2024; Cheng et al., 2022; Arvaniti & Claassen, 2017). There are no un/self-supervised methods that exist beyond manual gating and clustering to represent cytometry set data as a compact vector, therefore these stand as state-of-the-art for set representation.

Recent machine learning models designed for set data address permutation invariance and variable set sizes inherent in single-cell datasets. **Deep Sets** (Zaheer et al., 2017) made foundational contributions to set representation learning by introducing a permutation-invariant architecture capable of handling unordered sets. They established a universal approximation theorem for set functions, demonstrating that any permutation-invariant function can be represented by applying a function to each element individually followed by an aggregation step, such as summation. The **Set Transformer** (Lee et al., 2019) incorporates attention mechanisms designed to process unordered sets. Empirically, we find that size is still too large and other mechanisms must also be incorporated alongside ISAB. The **Optimal Transport Kernel Embedding** (OTKE) (Mialon et al., 2022) introduces a novel approach by aligning set representations through optimal transport distances while embedding them into a feature space using kernel techniques. OTKE ensures permutation invariance by treating sets as distributions, enabling robust and meaningful comparisons regardless of element order. The **Differentiable Expectation-Maximization** (DIEM) (Kim, 2022) integrates the Expectation-Maximization (EM) algorithm into a differentiable framework, enabling end-to-end training of neural networks on set-structured data. Although the previously mentioned methods lay the foundational framework for MAESTRO, they have not been applied to cytometry data and present limitations. Deep Sets and Set Transformer are supervised approaches, restricting their use to labeled datasets. Deep Sets and Set Transformer struggle to manage the number of cells in cytometry data. OTKE requires fixed-size input sets and similarly cannot handle the size of cytometry data. Lastly, DIEM is constrained by the absence of a public codebase.

In self-supervised learning, methods like **Masked Autoencoders (MAE)**, **DINO**, and **iBOT** have achieved remarkable success, particularly in computer vision tasks (He et al., 2022; Caron et al., 2021; Zhou et al., 2022). MAE learns representations by reconstructing masked portions of input data, capturing essential features without relying on explicit labels. DINO employs a student-teacher framework for self-distillation, where the student model learns from the teacher's outputs to achieve high performance, enhancing the efficiency of representation learning. iBOT extends these ideas by introducing an online tokenizer, facilitating the learning of meaningful token-based representations. By drawing on the strengths of these frameworks, MAESTRO addresses the unique challenges of set representation learning in cytometry data.

## 3 METHOD

Let $\mathcal{S}$ be a set in the form of a matrix of cytometry data (where rows are single cells and columns are features of cells). We aim to learn a function from the set of all finite sets of cells to $\mathbb{R}^D$, mapping variable-sized input sets to fixed-dimensional representations. Standard transformers are unsuitable for unordered sets due to their reliance on positional information. To overcome this, we employ specialized attention blocks lacking positional encodings, ensuring permutation invariance and accommodating variable cardinality. The initial layers use *permutation equivariant* operations to compute attention and extract features elementwise, allowing the attention mechanism to adjust appropriately with any input order. Finally, we apply pooling by attention to aggregate set information into a permutation invariant output. We take advantage of an online tokenizer teacher model to provide an encoded target of the full set of cells for a student model that encodes a masked subset of cells, thereby learning a representation of all cells in a set.

### 3.1 PRELIMINARIES - TRANSFORMER ATTENTION BLOCKS

#### 3.1.1 MULTI-HEAD ATTENTION BLOCK

The core component of our model is the Multi-Head Attention (MHA) mechanism (Vaswani et al., 2017), which enables the model to attend to information from multiple representation subspaces simultaneously. Given an input set $\mathcal{S} = \{x_1, x_2, \ldots, x_n\}$ where each $x_i \in \mathbb{R}^d$, we first project $\mathcal{S}$ into queries Q, keys K, and values V using learnable weight matrices $W_Q, W_K$, and $W_V$ respectively: $Q = \mathcal{S}W_Q, K = \mathcal{S}W_K$, and $V = \mathcal{S}W_V$. The attention mechanism for a single head is defined as

$$\text{Attention}(Q, K, V) = \text{softmax}\left(\frac{QK^\top}{\sqrt{d_k}}\right) V, \text{ where } d_k = \text{dimensionality of K} \tag{1}$$

To capture diverse features, MHA employs $H$ parallel attention heads, each computing its own attention output: $\text{head}_h = \text{Attention}(Q_h, K_h, V_h)$ for $h \in \{1, \ldots, H\}$. These outputs are concatenated and projected through a linear transformation with weight matrix $W_O \in \mathbb{R}^{Hd_v \times d}$:

$$\text{MHA}(Q, K, V) = \text{Concat}(\text{head}_1, \ldots, \text{head}_H)W_O, \text{ where } \text{head}_j = \text{Attention}(Q_j, K_j, V_j) \tag{2}$$

The Multi-Head Attention Block (MAB) integrates the MHA with additional components such as a cell level feed-forward neural network denoted by FF to enhance cell-level representations. Specifically, given set $S_0$ and set $S_1$, the MAB is defined as:

$$\text{MAB}(\mathcal{S}_0, \mathcal{S}_1) = \text{LayerNorm}\left(H + \text{FF}(H)\right) \tag{3}$$

where,

$$H = \text{LayerNorm}\left(\mathcal{S}_0 + \text{MHA}(\mathcal{S}_0, \mathcal{S}_1, \mathcal{S}_1)\right) \tag{4}$$

**Permutation Properties of MHA.** The Multi-Head Attention (MHA) mechanism exhibits different permutation properties depending on which of its inputs are permuted, crucial for understanding how our model handles shuffled input data.

**Theorem 1** (Permutation Properties of MHA). *Let $Q, K, V$ be the query, key, and value matrices respectively, and $P_Q, P_{KV}$ be permutation matrices. The Multi-Head Attention mechanism satisfies:*

$$MHA(P_Q Q, P_{KV} K, P_{KV} V) = P_Q \, MHA(Q, K, V).$$

*Proof.* See Appendix B.1. □

### 3.1.2 INDUCED SET-ATTENTION BLOCK

The **Induced Set-Attention Block (ISAB)** extends the MAB to efficiently handle large input sets. Based on our own studies, ISAB's alone are insufficient for the number of cells in cytometry data. Given an input set $\mathcal{S} \in \mathbb{R}^{n \times d}$ and a set of learnable inducing points $I \in \mathbb{R}^{m \times d}$, where $n$ is the number of elements in a set, $m$ is the number of inducing points, and $m << n$, the ISAB is formally defined as a two-step process:

$$\text{ISAB}_m(\mathcal{S}) = \text{MAB}(\mathcal{S}, H) \in \mathbb{R}^{n \times d}, \text{ where } H = \text{MAB}(I, \mathcal{S}) \in \mathbb{R}^{m \times d} \tag{5}$$

The first step transforms $I$ into $H$ by attending to the input set $\mathcal{S}$. The second step uses $H$, which encodes information about $\mathcal{S}$, to produce the final output. This structure reduces the computational complexity from $\mathcal{O}(n^2)$ to $\mathcal{O}(nm)$, where $n$ is the size of the input set. This formulation enables efficient processing of large sets assisting in the processing of large cytometry data sets. We find that ISAB's are also permutation equivariant:

**Theorem 2** (Permutation Equivariance of ISAB). *The Induced Set-Attention Block $ISAB_m(\mathcal{S})$ is permutation equivariant. Formally, for any permutation matrix $P$, it holds that:*

$$ISAB_m(P\mathcal{S}) = P\,ISAB_m(\mathcal{S}).$$

*Proof.* See Appendix B.2. □

### 3.1.3 POOLING BY MULTIHEAD ATTENTION

The **Pooling by Multihead Attention (PMA)** block provides a learnable aggregation mechanism for set-structured data. Given a set of input elements $\mathcal{S} = \{\mathbf{x}_1, \mathbf{x}_2, \ldots, \mathbf{x}_n\}$ where each $\mathbf{x}_i \in \mathbb{R}^d$, and a learnable token $\mathbf{s} \in \mathbb{R}^d$, the PMA is formally defined as:

$$\text{PMA}(\mathcal{S}) = \text{MAB}(\mathbf{s}, \mathcal{S}) \tag{6}$$

This attention-based pooling allows the model to assign varying importance to different set elements during aggregation, which is beneficial for identifying and focusing on the most relevant cell populations or features in cytometry data. This module is crucial to the permutation invariance property of MAESTRO, allowing for the collapse of hundreds of thousands of cells into a single, learnable token.

**Theorem 3** (Permutation Invariance of PMA). *The Pooling by Multihead Attention operation $PMA(\mathcal{S})$ is permutation invariant. Formally, for any permutation $\pi$ of the input set $\mathcal{S}$, it holds that:*

$$PMA(\pi(\mathcal{S})) = PMA(\mathcal{S})$$

*Proof.* See Appendix B.3. □

### 3.1.4 SELF-ATTENTION BLOCK

The **Self-Attention Block (SAB)** is a specialization of the MAB where the input set attends to itself. Given an input set $\mathcal{S} = \{x_1, x_2, \ldots, x_n\}$ where each $x_i \in \mathbb{R}^d$, the SAB is formally defined as:

$$\text{SAB}(\mathcal{S}) := \text{MAB}(\mathcal{S}, \mathcal{S}) \tag{7}$$

The SAB performs self-attention among the elements in the set, attending to itself, resulting in an output set of the same size as the input. This operation captures pairwise interactions between set elements, and stacking multiple SABs allows the model to encode higher-order interactions.

**Theorem 4** (Permutation Equivariance of SAB). *The Self-Attention Block $SAB(\mathcal{S})$ is permutation equivariant. Formally, for any permutation matrix $P$, it holds that:*

$$SAB(P\mathcal{S}) = P\,SAB(\mathcal{S}).$$

*Proof.* See Appendix B.4. □

## 3.2 MAESTRO ARCHITECTURE

### 3.2.1 NON-RANDOM BLOCK MASKING

We introduce Non-Random Block Masking (NRBM), a technique designed to enhance the model's learning of meaningful patterns by considering semantic relationships within the input data. Given an input set $\mathcal{S} = \{\mathbf{x}_1, \ldots, \mathbf{x}_n\}$ with $\mathbf{x}_i \in \mathbb{R}^d$ and a desired mask ratio $\rho \in [0, 1]$, NRBM first randomly selects a reference element $\mathbf{r}$ from $\mathcal{S}$. It then computes cosine similarities between $\mathbf{r}$ and all other elements, sorting the elements based on these similarities. This ordering ensures that semantically related elements are grouped together. A block mask is then created, with $\lfloor n\rho \rfloor$ masked elements in contiguous

---

**Algorithm 1** Non-Random Block Masking (NRBM)

**Input:** Set $\mathcal{S} = \{\mathbf{x}_1, \ldots, \mathbf{x}_n\} \subset \mathbb{R}^d$, mask ratio $\rho \in [0, 1]$

**Output:** Masked set $\mathcal{S}'$, mask vector $\mathbf{M}$

1. Select random reference $\mathbf{x}_r$ from $\mathcal{S}$
2. Compute similarities: $s_i = \text{sim}(\mathbf{x}_i, \mathbf{x}_r)$ for all $\mathbf{x}_i \in \mathcal{S}$
3. Sort $\mathcal{S}$ by descending $s_i$ to get $\mathcal{S}_{\text{sorted}}$
4. Compute mask size: $k = \lfloor \rho n \rfloor$
5. Create $\mathcal{S}'$ by masking $k$ contiguous elements in $\mathcal{S}_{\text{sorted}}$
6. Shuffle $\mathcal{S}'$
7. Generate mask vector $\mathbf{M}$ corresponding to $\mathcal{S}'$

**Return:** $\mathcal{S}'$, $\mathbf{M}$

---

blocks within this ordered set. The mask is applied by replacing masked elements with a special token $\mathbf{t}$. Finally, to maintain the model's permutation invariance, the resulting set is randomly shuffled. This approach destroys uninformative relationships due to redundant elements, challenging the model to infer broader patterns and ensures the model doesn't rely on any particular ordering of the input. We outline NRBM in Algorithm 0. For a detailed formulation of the NRBM process, including the specific steps and equations, please refer to Appendix C.1.

### 3.2.2 ENCODER AND DECODER

The encoder $f$ is defined as a series of nested operations, starting with a linear layer, followed by three consecutive ISABs (3.1.2), and ending with a PMA (3.1.3) operation:

$$f(\mathcal{S}) = \text{PMA}(\text{ISAB}_2(\text{ISAB}_1(\text{ISAB}_0(\text{Linear}(\mathcal{S}))))) = \mathbf{z} \tag{8}$$

where $\mathbf{z} = f(\mathcal{S}) \in \mathbb{R}^D$ is the latent representation.

The decoder $g$ takes the latent representation $\mathbf{z}$ and applies a PMA (3.1.3) to unpool the vector $z$ to the size of $\mathcal{S}$, this is followed by three consecutive SAB (3.1.4) operations and a linear layer to reduce the dimensionality back to the original size.

$$g(\mathbf{z}) = \text{Linear}(\text{SAB}_2(\text{SAB}_1(\text{SAB}_0(\text{PMA}(\mathbf{z}))))) = \hat{\mathcal{S}} \tag{9}$$

where $\hat{\mathcal{S}}$ is the reconstructed set.

**Reconstruction Loss** We employ a method called Sinkhorn Optimal Transport (SOT) for reconstruction loss (Cuturi). This approach allows us to compare the original and reconstructed sets without relying on a specific ordering of elements. We can think of this method as finding the best way to "match" elements from the original set to elements in the reconstructed set. The Sinkhorn-Knopp algorithm iteratively calculates this matching. Algorithm 2 describes the distance calculation from input to output. We define our reconstruction loss using the Sinkhorn Distance:

$$\mathcal{L}_{\text{rec}} = \text{SinkhornDistance}(\mathcal{S}, \hat{\mathcal{S}}) \tag{10}$$

where $\mathcal{S} = \{\mathbf{x}_1, ..., \mathbf{x}_n\}$ is the original set and $\hat{\mathcal{S}} = \{\hat{\mathbf{x}}_1, ..., \hat{\mathbf{x}}_n\}$ is the reconstructed set.

---

**Algorithm 2** Sinkhorn Optimal Transport Distance

**Input:** Sets $\mathcal{S}, \hat{\mathcal{S}}$, Iterations $T$

**Output:** Sinkhorn Distance $d$

1. Compute pairwise distance matrix $D_{ij} = \text{distance}(\mathbf{x}_i \in \mathcal{S}, \hat{\mathbf{x}}_j \in \hat{\mathcal{S}})$
2. Initialize $A_{ij} = \frac{1}{n}$ for all $i, j$, where $A$ is an $n \times n$ matrix
3. **for** $t = 1$ to $T$ **do**
4.    Normalize each row of $A$ such that $\sum_j A_{ij} = 1$
5.    Normalize each column of $A$ such that $\sum_i A_{ij} = 1$
6.    Update $A_{ij} = A_{ij} \cdot e^{-D_{ij}}$
7. Compute $d = \sum_{i,j} A_{ij} \cdot D_{ij}$

**Return:** $d$

---

This loss function ensures that the reconstruction quality is evaluated in a permutation-invariant manner, as it only depends on how well we can match the original and reconstructed sets, not their order. For a more detailed mathematical formulation of the SOT and the reconstruction loss, refer to Appendix C.2.

### 3.2.3 ONLINE TOKENIZER VIA SELF-DISTILLATION

MAESTRO employs a self-distillation framework with an online tokenizer to tackle the computational challenges of encoding large cytometry datasets. In this context, the teacher model acts as an online tokenizer, providing a token distribution target of the full set of cells (elements) for the student model, which operates on a masked subset. Let $\mathcal{S} = \{\mathbf{x}_1, \mathbf{x}_2, \ldots, \mathbf{x}_n\}$ be the input set of cells, where each $\mathbf{x}_i \in \mathbb{R}^d$ represents a cell's high-dimensional measurement. We define the student model $f_s$ and the teacher model $f_t$ using the encoder structure:

$$f_s(\mathcal{S}_m) = f(\text{NRBM}(\mathcal{S}_m), \rho) \tag{11}$$

$$f_t(\mathcal{S}) = f(\mathcal{S}) \tag{12}$$

where $\rho$ is the masking rate and $\mathcal{S}_m \subset \mathcal{S}$ is a subset of size $m \ll n$.

The teacher's parameters $\theta_t$ are an exponential moving average (EMA) of the student's parameters $\theta_s$:

$$\theta_t = \alpha\theta_t + (1 - \alpha)\theta_s \tag{13}$$

where $\alpha$ is the EMA decay rate. The online tokenizer (teacher) provides a stable target for the student model. The self-distillation process is achieved by minimizing the KL divergence between the softmax distributions of the latent representations produced by the student and teacher models:

$$\mathcal{L}_{\text{SD}} = \frac{1}{m} \sum_{i=1}^{m} \text{KL}(\text{softmax}(f_s(\mathbf{x}_i)/\tau) \, \| \, \text{softmax}(f_t(\mathbf{x}_i)/\tau)) \tag{14}$$

where $\tau$ is a temperature parameter that controls the softness of the probability distributions, and $\mathbf{x}_i \in \mathcal{S}_m$ are the elements of the masked subset. This loss encourages the student model to produce latent representations that are similar to those of the teacher model, effectively distilling the knowledge from the full dataset into the student model that only sees a masked subset. Notably, this approach is feasible because the teacher model, operating on the full set, does not require back-propagation or gradient calculations, which would be computationally prohibitive for large datasets. The student model, working on a smaller subset, can perform gradient calculations and back-propagation, enabling it to learn and update its parameters.

### 3.2.4 OVERALL

Using all of the above components, we describe the MAESTRO algorithm here:

---

**Algorithm 3** MAESTRO Model Overview

---

**Input:** Input set $\mathcal{S}$, mask ratio $\rho$, EMA decay rate $\alpha$, temperature $\tau$
**Output:** Reconstructed set $\hat{\mathcal{S}}$

1. **Initialize** student parameters $\theta_s$ and teacher parameters $\theta_t \leftarrow \theta_s$
2. **Randomly sample** subset $\mathcal{S}_M \subset \mathcal{S}$
3. **Apply NRBM** to $\mathcal{S}_M$ to obtain masked set $\mathcal{S}'$ and mask vector $\mathbf{M}$: $(\mathcal{S}', \mathbf{M}) = \text{NRBM}(\mathcal{S}_M, \rho)$
4. **Encode masked set** using student encoder: $\mathbf{z}_s = f_s(\mathcal{S}'; \theta_s)$
5. **Decode latent representation** to reconstruct the set: $\hat{\mathcal{S}} = g(\mathbf{z}_s; \theta_s)$
6. **Encode original set** using teacher encoder: $\mathbf{z}_t = f_t(\mathcal{S}; \theta_t)$
7. **Compute reconstruction loss**: $\mathcal{L}_{\text{rec}} = \text{SinkhornDistance}(\mathcal{S}_M, \hat{\mathcal{S}})$
8. **Compute self-distillation loss**: $\mathcal{L}_{\text{SD}} = \frac{1}{|\mathcal{S}_M|} \sum_{i=1}^{|\mathcal{S}_M|} \text{KL}\left(\text{softmax}\left(\frac{\mathbf{z}_s^i}{\tau}\right) \, \middle\| \, \text{softmax}\left(\frac{\mathbf{z}_t^i}{\tau}\right)\right)$
9. **Compute total loss**: $\mathcal{L} = \mathcal{L}_{\text{rec}} + \mathcal{L}_{\text{SD}}$
10. **Update student parameters**: $\theta_s \leftarrow \theta_s - \eta \nabla_{\theta_s} \mathcal{L}$
11. **Update teacher parameters** using EMA: $\theta_t \leftarrow \alpha \theta_t + (1 - \alpha) \theta_s$
12. **Return** reconstructed set $\hat{\mathcal{S}}$

---

## 4 EXPERIMENT

To evaluate the effectiveness of MAESTRO in representing and analyzing cytometry set data, we conducted experiments using a large cohort of cytometry samples. We assessed the model's performance in reconstructing masked cells, latent embedding representativeness, and linear probing tasks to evaluate the effectiveness of embedding important immunological concepts in the representation.

### 4.1 DATA

We utilized a dataset of 1,514 whole blood cytometry samples spanning 14 cohorts and 11 phenotypes (Appendix E.1). Data were generated at three locations over various time points (Appendix E.2), with raw data showing batch effects (Appendix E.3.1). We employed the technical control sample BatchControlHD2, which exhibits minimal batch effects in learned representations (Figure 3) compared to raw data (Appendix E.3.2). Disease diagnostic and meta data were provided by the primary clinician teams for each study. Appendix E.1 displays the distribution of cell counts per sample (minimum=11,829; maximum=1,386,520), highlighting dataset variability. Each sample is represented as a matrix with cells as rows and proteins as columns. Notably, cell types obtained through manual gating are used only to evaluate the representations learned by MAESTRO.

### 4.2 PRE-TRAINING: RECONSTRUCTION OF MASKED CELLS

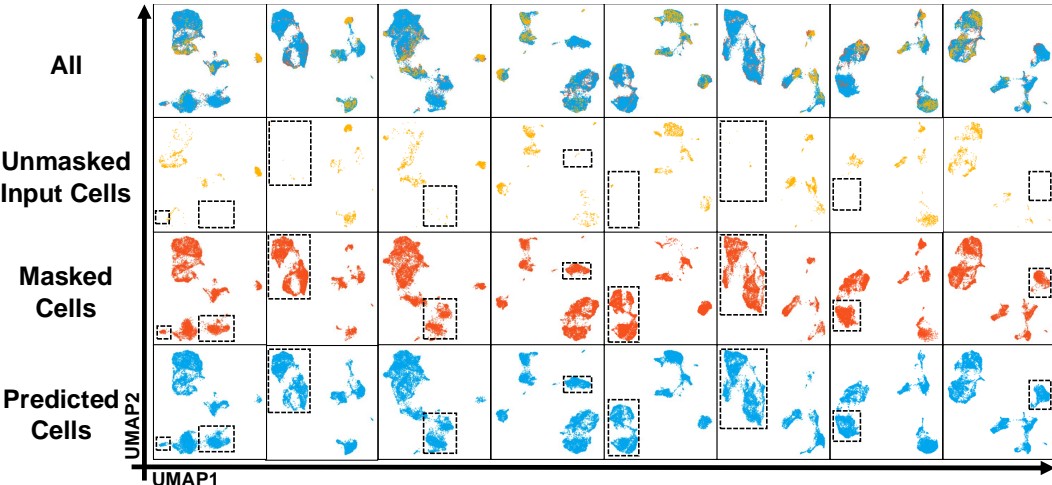

Figure 2: **Set reconstruction.** UMAP of the unmasked input cells (yellow, row two), the original masked cells (red, row three), and the predicted cells (blue, row 4). The top row shows all of these overlaid. Each column represents a random held-out test sample. Dotted boxes highlight areas where very few or no cells exist in the unmasked input set but are still reconstructed by MAESTRO.

To demonstrate MAESTRO's capability in accurately reconstructing masked cells, one of the objectives during training, Figure 2 showcases the reconstruction results for eight randomly selected samples from a held-out test set during pre-training. In these plots, the unmasked input cells are depicted in yellow (row two), the original masked cells in red (row three), and the predicted reconstructions by MAESTRO are shown in blue (row four). The visualization shows a projection of the cells onto a two-dimensional space using Uniform Manifold Approximation and Projection (UMAP) (McInnes et al., 2022). The close alignment between the original and reconstructed cells indicates that MAESTRO effectively captures the underlying structure of the data. Due to the NRBM procedure, there are certain regional islands (as denoted by black, dashed boxes) of the UMAP where few or no unmasked input cells exist, but are still accurately reconstructed. This demonstrates the model's ability to infer complex patterns and relationships within the data, even for cells that are distinct from other input populations. Pre-training details and implementation including runtime and memory usage can be found in Appendix F.2.

## 4.3 VECTORIZED IMMUNE PROFILES

We evaluated the quality of MAESTRO's latent representations by projecting the 1,024-dimensional embeddings into two dimensions using UMAP for visualization, colored by each sample's diagnosis (Figure 3). The resulting plot shows that samples cluster by diagnosis, indicating that MAESTRO effectively captures meaningful patterns related to disease phenotypes. Samples denoted as BatchControlHD2 are technical controls used in our data generation process, which we see cluster among themselves despite exhibiting batch effects in the raw data E.3.2. We quantitatively assessed the representations by generating a contingency table of nearest neighbor diagnosis proportions (Figure 3). This reveals that many samples have nearest neighbors with matching diagnoses, as shown by the high values along the diagonal, supporting the model's ability to represent meaningful information.

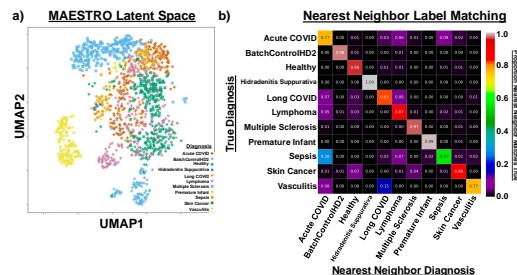

Figure 3: **Vectorized immune profiles**. a) Sample-level (set) latent embeddings projected into a low-dimensional space. Each diagnosis has a unique shape-color combination for its point. Samples tend to cluster by diagnosis. b) The y-axis is true diagnosis of each point, the x-axis is the diagnosis of each point's nearest neighbor. We show the proportion of nearest neighbor diagnosis to each point's true diagnosis. High values along the diagonal demonstrate the majority of samples have a nearest neighbor with matching diagnosis.

## 4.4 LINEAR PROBING

To evaluate the discriminative power of the learned representations, we performed a linear probing task where we used the latent representations as inputs to a basic regression model for predicting diagnosis, age, and sex. Figure 4 summarizes the performance of MAESTRO compared to other methods (see Appendix D for tabular performance metrics). We compare MAESTRO to manual gating and clustering which are the gold standard for sample representations in biomedical research due to the lack of set representation methods that can handle the idiosyncrasies of cytometry data. We also provide benchmarks against previously mentioned methods (Deep Sets, Set Transformer, and OTKE) that have never been used on cytometry data before and are suitable for set analysis. These methods are unable to handle the number of cells in a sample or require a fixed size input (further highlighting MAESTRO's capabilities), so we have a random subset of 10,000 cells as input for these models. Details on implementations for each of these methods can be found in Appendix F.3. MAESTRO achieved superior accuracy, area under receiver operator curve, and F1 score in classifying diagnosis and sex, and achieved the lowest MAE and highest R-squared values for age regression. This indicates that the representations learned by our model are more informative to downstream prediction tasks. Lastly, we perform an ablation study on the same task using our MAESTRO architecture, indicating the importance of various design choices (Table 1). The proposed MAESTRO model outperforms other variants of the same model.

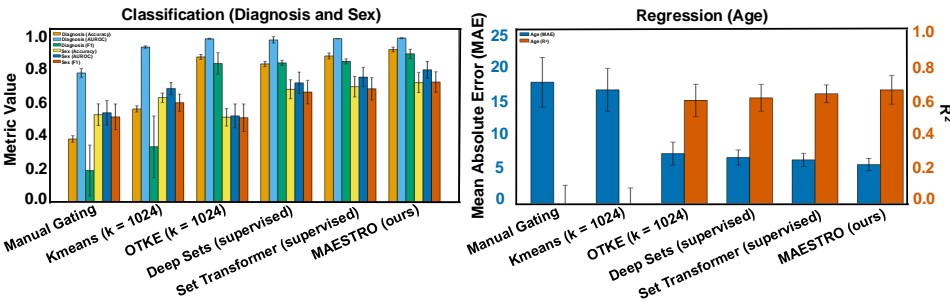

Figure 4: **Linear probe.** Classification and regression benchmark of sample diagnosis, sex, and age. MAESTRO outperforms all benchmarked methods across all metrics. Our study provides benchmarks against OTKE, DeepSets, and Set Transformer. These methods are suitable for set representation learning but have never been applied to cytometry.

| Model | Accuracy | AUROC | F1 |
|---|---|---|---|
| Random Masking | $0.887 \pm 0.029$ | $0.987 \pm 0.007$ | $0.837 \pm 0.052$ |
| + Multi-rate Masking | $0.898 \pm 0.023$ | $0.987 \pm 0.004$ | $0.861 \pm 0.036$ |
| + Block Masking | $0.909 \pm 0.025$ | $0.991 \pm 0.004$ | $0.873 \pm 0.042$ |
| + Self-Distillation (full model) | $\mathbf{0.923 \pm 0.014}$ | $\mathbf{0.992 \pm 0.003}$ | $\mathbf{0.897 \pm 0.029}$ |
| - Random Masking | $0.721 \pm 0.023$ | $0.955 \pm 0.014$ | $0.485 \pm 0.029$ |

Table 1: **Ablation study.** Each subsequent row builds upon the module of previous rows above denoted by +/-. Notably, the masked modelling and self-distillation modules are imperative aspects to the performance of the model. The removal of masked modelling has a major impact on the performance of the model. Details of ablation implementation can be found in Appendix F.4.

## 4.5 RETRIEVAL OF CELL-TYPE DISTRIBUTIONS

Many SSL studies that operate at the cell-level aim to predict cell-type or use it as a metric for evaluation (Cui et al., 2024; Kim et al., 2024). Our study operates at the sample-level, therefore we predict entire cell-type distributions, a much harder task. Accurately predicting cell-type distributions of samples is of significant biological importance, as it can provide insight into the immune landscape and disease mechanisms. We assessed MAESTRO's ability to retrieve these distributions by using the latent representations as input to a single-layer neural network to predict known cell-type distributions obtained through traditional manual gating. Figure 5 benchmarks MAESTRO's performance in predicting cell-type distributions averaged across all cells (Figure 5a) and against 16 individual cell-types (Figure 5b). We benchmark MAESTRO against k-means clustering, OTKE, Deep Sets, and Set Transformer. Details on each implementation can be found in

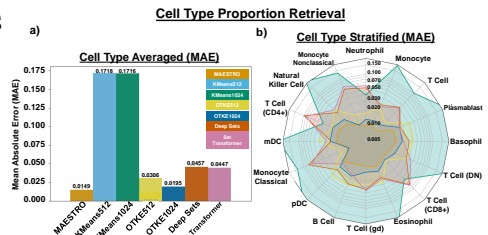

Figure 5: **Cell-type distribution retrieval**. Color scheme is shared between plot a and b. a) From each sample representation we predict the distribution of 16 various cell-types. We average across all 16 cell-types to show general performance by mean absolute error (MAE). b) Radar plot showing the performance of each model in predicting 16 different cell-types. Each line-color represents a different model, each cell-type is represented by a spoke, values denote MAE. As we move away from the center the MAE gets higher, less coverage across the map means lower error.

F.5. MAESTRO outperforms all benchmarks methods, suggesting that MAESTRO not only captures global patterns pertinent to diagnosis, sex, and age but also encodes local, cell-level information critical for understanding cellular heterogeneity.

## 5 CONCLUSION

MAESTRO is a learning architecture that generates representations of immune profiles from high-dimensional cytometry data — a clinically relevant modality due to its direct measurement of protein expression on immune cells, providing more immediate functional insights. MAESTRO effectively handles variable set sizes, permutation invariance, and computational constraints in single-cell datasets. It produces holistic sample representations that contain immunologically meaningful information, complementing the standard manual gating approach, as demonstrated by probe experiments evaluating age, sex, and strong diagnosis predictions, highlighting their clinical and immunological relevance. Additionally, the ability to infer cell type distributions from embeddings shows that the embeddings contain biologically meaningful information. MAESTRO's holistic representation of the immune system is essential for predicting outcomes, identifying health trajectories, and advancing precision medicine—capabilities that are difficult to achieve with previous approaches. Further potential applications include treatment response prediction, early disease detection, differentiating vaccine response across diseases, assessing patient protection levels from vaccines at specific time-points, and disease subtyping. Limitations, such as single-sample batch processing and challenges in detecting rare populations, are discussed in Appendix F.6. Future applications include adapting MAESTRO to other single-cell modalities, computational histopathology, and multi-modal representations to further advance biomedical and immunological discoveries.

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

## A  ACKNOWLEDGMENTS

We would like to acknowledge that Figure 1 was created with BioRender.com. Further, we acknowledge that Large Language Models (LLMs) were used throughout the study.

# B  PROOFS

## B.1  PROOF OF MHA PERMUTATION PROPERTIES

We need to prove three properties of Multi-Head Attention (MHA):

1. Permutation Equivariance: When all inputs are permuted. 2. Permutation Equivariance for Q: When only the query is permuted. 3. Permutation Invariance for K and V: When only keys and values are permuted.

*Proof.* Let $\mathbf{Q} \in \mathbb{R}^{n_q \times d}$, $\mathbf{K}, \mathbf{V} \in \mathbb{R}^{n_{kv} \times d}$ be the query, key, and value matrices respectively, and $\mathbf{P}_q \in \mathbb{R}^{n_q \times n_q}$, $\mathbf{P}_{kv} \in \mathbb{R}^{n_{kv} \times n_{kv}}$ be permutation matrices.

1. Permutation Equivariance (all inputs): When $n_q = n_{kv}$ and $\mathbf{P} = \mathbf{P}_q = \mathbf{P}_{kv}$:

$$
\begin{aligned}
\mathrm{MHA}(\mathbf{PQ}, \mathbf{PK}, \mathbf{PV}) &= \mathrm{softmax}\left(\frac{(\mathbf{PQ})(\mathbf{PK})^\top}{\sqrt{d_k}}\right)\mathbf{PV} \\
&= \mathrm{softmax}\left(\frac{\mathbf{PQK}^\top\mathbf{P}^\top}{\sqrt{d_k}}\right)\mathbf{PV} \\
&= \mathbf{P}\,\mathrm{softmax}\left(\frac{\mathbf{QK}^\top}{\sqrt{d_k}}\right)\mathbf{V} \\
&= \mathbf{P}\,\mathrm{MHA}(\mathbf{Q}, \mathbf{K}, \mathbf{V})
\end{aligned}
$$

2. Permutation Equivariance for Q: When only Q is permuted:

$$
\begin{aligned}
\mathrm{MHA}(\mathbf{P}_q\mathbf{Q}, \mathbf{K}, \mathbf{V}) &= \mathrm{softmax}\left(\frac{(\mathbf{P}_q\mathbf{Q})\mathbf{K}^\top}{\sqrt{d_k}}\right)\mathbf{V} \\
&= \mathrm{softmax}\left(\frac{\mathbf{P}_q\mathbf{QK}^\top}{\sqrt{d_k}}\right)\mathbf{V} \\
&= \mathbf{P}_q\,\mathrm{softmax}\left(\frac{\mathbf{QK}^\top}{\sqrt{d_k}}\right)\mathbf{V} \\
&= \mathbf{P}_q\,\mathrm{MHA}(\mathbf{Q}, \mathbf{K}, \mathbf{V})
\end{aligned}
$$

3. Permutation Invariance for K and V: When only K and V are permuted:

$$
\begin{aligned}
\mathrm{MHA}(\mathbf{Q}, \mathbf{P}_{kv}\mathbf{K}, \mathbf{P}_{kv}\mathbf{V}) &= \mathrm{softmax}\left(\frac{\mathbf{Q}(\mathbf{P}_{kv}\mathbf{K})^\top}{\sqrt{d_k}}\right)\mathbf{P}_{kv}\mathbf{V} \\
&= \mathrm{softmax}\left(\frac{\mathbf{QK}^\top\mathbf{P}_{kv}^\top}{\sqrt{d_k}}\right)\mathbf{P}_{kv}\mathbf{V} \\
&= \mathrm{softmax}\left(\frac{\mathbf{QK}^\top}{\sqrt{d_k}}\right)\mathbf{V} \\
&= \mathrm{MHA}(\mathbf{Q}, \mathbf{K}, \mathbf{V})
\end{aligned}
$$

These properties together demonstrate that: a) MHA is permutation equivariant when all inputs are permuted equally. b) MHA is permutation equivariant with respect to Q when only Q is permuted. c) MHA is permutation invariant with respect to K and V when only K and V are permuted.

□

## B.2 PROOF OF ISAB PERMUTATION EQUIVARIANCE

To prove that the Induced Set-Attention Block (ISAB) is permutation equivariant, we need to show that for any permutation matrix $\mathbf{P}$, $\text{ISAB}_m(\mathbf{P}\mathcal{S}) = \mathbf{P}\,\text{ISAB}_m(\mathcal{S})$.

*Proof.* Let $\mathcal{S} \in \mathbb{R}^{n \times d}$ be the input set, $\mathbf{I} \in \mathbb{R}^{m \times d}$ be the set of learnable inducing points, and $\mathbf{P} \in \mathbb{R}^{n \times n}$ be any permutation matrix.

Recall that ISAB is defined as a two-step process:

$$\mathbf{H} = \text{MAB}(\mathbf{I}, \mathcal{S}, \mathcal{S})$$
$$\text{ISAB}_m(\mathcal{S}) = \text{MAB}(\mathcal{S}, \mathbf{H}, \mathbf{H})$$

1) First, we examine how permutation affects $\mathbf{H}$:

$$\mathbf{H}' = \text{MAB}(\mathbf{I}, \mathbf{P}\mathcal{S}, \mathbf{P}\mathcal{S})$$
$$= \text{LayerNorm}(\mathbf{I} + \text{MHA}(\mathbf{I}, \mathbf{P}\mathcal{S}, \mathbf{P}\mathcal{S}))$$

From our above MHA proof, we know that MHA is invariant when only K and V are permuted. Therefore:

$$\mathbf{H}' = \mathbf{H}$$

2) Now, we examine the second step:

$$\text{ISAB}_m(\mathbf{P}\mathcal{S}) = \text{MAB}(\mathbf{P}\mathcal{S}, \mathbf{H}, \mathbf{H})$$
$$= \text{LayerNorm}(\mathbf{P}\mathcal{S} + \text{MHA}(\mathbf{P}\mathcal{S}, \mathbf{H}, \mathbf{H}))$$

3) To show equivariance, we need to prove that this is equal to $\mathbf{P}\text{ISAB}_m(\mathcal{S})$:

$$\mathbf{P}\text{ISAB}_m(\mathcal{S}) = \mathbf{P}\text{MAB}(\mathcal{S}, \mathbf{H}, \mathbf{H})$$
$$= \mathbf{P}\text{LayerNorm}(\mathcal{S} + \text{MHA}(\mathcal{S}, \mathbf{H}, \mathbf{H}))$$

4) The equality holds because:

$$\text{LayerNorm}(\mathbf{P}\mathcal{S} + \text{MHA}(\mathbf{P}\mathcal{S}, \mathbf{H}, \mathbf{H})) = \text{LayerNorm}(\mathbf{P}\mathcal{S} + \mathbf{P}\text{MHA}(\mathcal{S}, \mathbf{H}, \mathbf{H}))$$
$$= \mathbf{P}\text{LayerNorm}(\mathcal{S} + \text{MHA}(\mathcal{S}, \mathbf{H}, \mathbf{H}))$$

This equality holds because: a) LayerNorm is permutation equivariant b) MHA is permutation equivariant in its first argument (as shown in the revised MHA proof) c) $\mathbf{H}$ remains unchanged under permutation of $\mathcal{S}$

Therefore, we have shown that $\text{ISAB}_m(\mathbf{P}\mathcal{S}) = \mathbf{P}\,\text{ISAB}_m(\mathcal{S})$ for any permutation matrix $\mathbf{P}$, proving that ISAB is permutation equivariant. $\square$

## B.3 PROOF OF PMA PERMUTATION INVARIANCE

To prove that the Pooling by Multihead Attention (PMA) operation is permutation invariant, we need to show that for any permutation matrix $\mathbf{P}$ applied to the input set $\mathcal{S}$, $\text{PMA}(\mathbf{P}\mathcal{S}) = \text{PMA}(\mathcal{S})$.

*Proof.* Let $\mathcal{S} \in \mathbb{R}^{n \times d}$ be the input set, $\mathbf{s} \in \mathbb{R}^{1 \times d}$ be the learnable token, and $\mathbf{P} \in \mathbb{R}^{n \times n}$ be any permutation matrix.

Recall that PMA is defined as:
$$\text{PMA}(\mathcal{S}) = \text{MAB}(\mathbf{s}, \mathcal{S}, \mathcal{S})$$

1) First, let's consider PMA applied to the permuted input:
$$\text{PMA}(\mathbf{P}\mathcal{S}) = \text{MAB}(\mathbf{s}, \mathbf{P}\mathcal{S}, \mathbf{P}\mathcal{S})$$

2) Expanding the MAB operation:
$$\text{MAB}(\mathbf{s}, \mathbf{P}\mathcal{S}, \mathbf{P}\mathcal{S}) = \text{LayerNorm}(\mathbf{s} + \text{MHA}(\mathbf{s}, \mathbf{P}\mathcal{S}, \mathbf{P}\mathcal{S}))$$

3) From our above MHA proof, we know that MHA is invariant when only K and V are permuted, which is the case here since $\mathbf{s}$ is not permuted. Therefore:
$$\text{MHA}(\mathbf{s}, \mathbf{P}\mathcal{S}, \mathbf{P}\mathcal{S}) = \text{MHA}(\mathbf{s}, \mathcal{S}, \mathcal{S})$$

4) Consequently:
$$\begin{aligned} \text{MAB}(\mathbf{s}, \mathbf{P}\mathcal{S}, \mathbf{P}\mathcal{S}) &= \text{LayerNorm}(\mathbf{s} + \text{MHA}(\mathbf{s}, \mathbf{P}\mathcal{S}, \mathbf{P}\mathcal{S})) \\ &= \text{LayerNorm}(\mathbf{s} + \text{MHA}(\mathbf{s}, \mathcal{S}, \mathcal{S})) \\ &= \text{MAB}(\mathbf{s}, \mathcal{S}, \mathcal{S}) \end{aligned}$$

5) This shows that:
$$\text{PMA}(\mathbf{P}\mathcal{S}) = \text{PMA}(\mathcal{S})$$

Therefore, we have shown that $\text{PMA}(\mathbf{P}\mathcal{S}) = \text{PMA}(\mathcal{S})$ for any permutation matrix $\mathbf{P}$, proving that PMA is permutation invariant.

$\square$

### B.4 PROOF OF SAB PERMUTATION EQUIVARIANCE

To prove that the Self-Attention Block (SAB) is permutation equivariant, we need to show that for any permutation matrix $\mathbf{P}$, $\text{SAB}(\mathbf{P}\mathcal{S}) = \mathbf{P}\,\text{SAB}(\mathcal{S})$.

*Proof.* Let $\mathcal{S} \in \mathbb{R}^{n \times d}$ be the input set and $\mathbf{P} \in \mathbb{R}^{n \times n}$ be any permutation matrix.

Recall that SAB is defined as:
$$\text{SAB}(\mathcal{S}) := \text{MAB}(\mathcal{S}, \mathcal{S}, \mathcal{S})$$

1) We start by expanding the MAB operation:
$$\begin{aligned}
\text{SAB}(\mathcal{S}) &= \text{MAB}(\mathcal{S}, \mathcal{S}, \mathcal{S}) \\
&= \text{LayerNorm}(H + \text{FF}(H)) \\
\text{where } H &= \text{LayerNorm}(\mathcal{S} + \text{MHA}(\mathcal{S}, \mathcal{S}, \mathcal{S}))
\end{aligned}$$

2) Now, let's apply a permutation $\mathbf{P}$ to the input:
$$\begin{aligned}
\text{SAB}(\mathbf{P}\mathcal{S}) &= \text{MAB}(\mathbf{P}\mathcal{S}, \mathbf{P}\mathcal{S}, \mathbf{P}\mathcal{S}) \\
&= \text{LayerNorm}(H' + \text{FF}(H')) \\
\text{where } H' &= \text{LayerNorm}(\mathbf{P}\mathcal{S} + \text{MHA}(\mathbf{P}\mathcal{S}, \mathbf{P}\mathcal{S}, \mathbf{P}\mathcal{S}))
\end{aligned}$$

3) Using the permutation equivariance of MHA when all inputs are permuted (proven earlier):
$$\begin{aligned}
H' &= \text{LayerNorm}(\mathbf{P}\mathcal{S} + \mathbf{P}\text{MHA}(\mathcal{S}, \mathcal{S}, \mathcal{S})) \\
&= \text{LayerNorm}(\mathbf{P}(\mathcal{S} + \text{MHA}(\mathcal{S}, \mathcal{S}, \mathcal{S}))) \\
&= \mathbf{P}\text{LayerNorm}(\mathcal{S} + \text{MHA}(\mathcal{S}, \mathcal{S}, \mathcal{S})) \\
&= \mathbf{P}H
\end{aligned}$$

4) The feedforward network FF is applied elementwise, so it's permutation equivariant:
$$\text{FF}(\mathbf{P}H) = \mathbf{P}\text{FF}(H)$$

5) Combining these results:
$$\begin{aligned}
\text{SAB}(\mathbf{P}\mathcal{S}) &= \text{LayerNorm}(\mathbf{P}H + \text{FF}(\mathbf{P}H)) \\
&= \text{LayerNorm}(\mathbf{P}H + \mathbf{P}\text{FF}(H)) \\
&= \text{LayerNorm}(\mathbf{P}(H + \text{FF}(H))) \\
&= \mathbf{P}\text{LayerNorm}(H + \text{FF}(H)) \\
&= \mathbf{P}\text{SAB}(\mathcal{S})
\end{aligned}$$

Therefore, we have shown that $\text{SAB}(\mathbf{P}\mathcal{S}) = \mathbf{P}\,\text{SAB}(\mathcal{S})$ for any permutation matrix $\mathbf{P}$, proving that SAB is permutation equivariant. $\qquad\square$

## C  FORMAL DEFINITIONS

### C.1  NON-RANDOM BLOCK MASKING

Let $\mathcal{S} = \{\mathbf{x}_1, \mathbf{x}_2, \ldots, \mathbf{x}_n\}$ be an input set where each $\mathbf{x}_i \in \mathbb{R}^d$, and let $\rho \in [0, 1]$ be the desired mask ratio. The Non-Random Block Masking (NRBM) process consists of the following steps:

1. **Random Element Selection:** Select a random element $\mathbf{r}$ from the input set:

$$\mathbf{r} = \mathbf{x}_k, \quad k \sim \text{Uniform}\{1, \ldots, n\}$$

This element serves as a reference for similarity calculations.

2. **Similarity Calculation:** Compute the cosine similarity between each element and the randomly selected element:

$$s_i = \text{cosine\_similarity}(\mathbf{x}_i, \mathbf{r}) = \frac{\mathbf{x}_i \cdot \mathbf{r}}{\|\mathbf{x}_i\|\|\mathbf{r}\|}, \quad \forall i \in \{1, \ldots, n\}$$

3. **Similarity-based Ordering:** Define a permutation $\pi : \{1, \ldots, n\} \to \{1, \ldots, n\}$ that sorts elements based on their similarity to $\mathbf{r}$:

$$s_{\pi(i)} \geq s_{\pi(j)} \quad \text{for all } i < j$$

This ordering ensures that similar elements are grouped together.

4. **Block Mask Creation:** Create a binary mask $\mathbf{M} = (M_1, \ldots, M_n)$ with the following parameters:

   - Number of elements to mask: $m = \lfloor n\rho \rfloor$
   - Block size: $b = \max(1, \lfloor \frac{m}{n/m} \rfloor)$

   Define the mask as:

$$M_i = \begin{cases} 1, & \text{if } \pi^{-1}(i) \bmod (2b) < b \text{ and } \sum_{j=1}^{i-1} M_j < m \\ 0, & \text{otherwise} \end{cases}$$

   This creates contiguous blocks of masked elements in the similarity-ordered sequence.

5. **Masked Set Creation:** Let $\mathbf{t} \in \mathbb{R}^d$ be the mask token. Create the masked set $\mathcal{S}'$ by replacing masked elements with $\mathbf{t}$:

$$\mathcal{S}' = \{\mathbf{y}_1, \mathbf{y}_2, \ldots, \mathbf{y}_n\}, \quad \text{where } \mathbf{y}_i = \begin{cases} \mathbf{t}, & \text{if } M_i = 1 \\ \mathbf{x}_i, & \text{if } M_i = 0 \end{cases}$$

The complete NRBM function is then defined as:

$$\text{NRBM} : \mathcal{P}(\mathbb{R}^d) \times [0, 1] \to \mathcal{P}(\mathbb{R}^d) \times \{0, 1\}^n \times \mathcal{S}_n$$

$$\text{NRBM}(\mathcal{S}, \rho) = (\mathcal{S}', \mathbf{M}, \pi)$$

where $\mathcal{S}'$ is the masked set, $\mathbf{M}$ is the binary mask, and $\pi$ is the permutation used for sorting.

This formulation ensures that:

- The masking is based on similarity to a randomly chosen element, promoting semantic coherence.
- Masks are applied in contiguous blocks, encouraging the model to learn robust patterns.
- The original ordering can be restored using the returned permutation $\pi$.

## C.2 SINKHORN OPTIMAL TRANSPORT

To ensure our reconstruction loss is permutation-invariant, we employ Sinkhorn Optimal Transport. This method allows us to compare the original and reconstructed sets without relying on a specific ordering of elements.

The key idea is to find an optimal 'matching" between the original set $\mathcal{S} = \{\mathbf{x}_1, ..., \mathbf{x}_n\}$ and the reconstructed set $\hat{\mathcal{S}} = \{\hat{\mathbf{x}}_1, ..., \hat{\mathbf{x}}_n\}$. This matching is represented by a transport plan $\mathbf{P}$.

The transport plan $\mathbf{P}$ is an $n \times n$ matrix where each element $P_{ij}$ represents how much of element $\mathbf{x}_i$ from the original set is 'transported" to element $\hat{\mathbf{x}}_j$ in the reconstructed set. Formally:

$$\mathbf{P} = [P_{ij}] \in \mathbb{R}^{n \times n}, \quad \text{where } 0 \leq P_{ij} \leq 1$$

To ensure that $\mathbf{P}$ represents a valid matching, we constrain it to be a doubly stochastic matrix:

$$\sum_{j=1}^{n} P_{ij} = 1 \quad \text{for all } i, \quad \text{and} \quad \sum_{i=1}^{n} P_{ij} = 1 \quad \text{for all } j$$

These constraints ensure that each original element is fully 'distributed" across the reconstructed elements, and each reconstructed element is fully 'accounted for" by the original elements.

We define a cost matrix $\mathbf{C}$, where each element $C_{ij}$ represents the 'cost" of transporting $\mathbf{x}_i$ to $\hat{\mathbf{x}}_j$:

$$C_{ij} = \|\mathbf{x}_i - \hat{\mathbf{x}}_j\|_2^2$$

The goal is to find the optimal transport plan $\mathbf{P}^*$ that minimizes the total transport cost:

$$\mathbf{P}^* = \arg\min_{\mathbf{P}} \sum_{i,j} P_{ij} C_{ij} \quad \text{subject to the constraints on } \mathbf{P}$$

To make this optimization problem tractable, we add an entropy regularization term:

$$\mathbf{P}^* = \arg\min_{\mathbf{P}} \sum_{i,j} P_{ij} C_{ij} - \epsilon \sum_{i,j} P_{ij} \log P_{ij}$$

where $\epsilon > 0$ is the regularization strength.

This problem can be solved efficiently using the Sinkhorn-Knopp algorithm, which iteratively updates $\mathbf{P}$ until convergence.

Finally, we define our reconstruction loss as:

$$\mathcal{L}_{\text{rec}} = \sum_{i,j} P_{ij}^* C_{ij}$$

This loss function ensures that the reconstruction quality is evaluated in a permutation-invariant manner, as it only depends on the optimal matching between the original and reconstructed sets, not their order.

# D  APPENDIX TABLES

## D.1  DIAGNOSIS

| Model | Accuracy | AUROC | F1 |
|---|---|---|---|
| Manual Gating | $0.381 \pm 0.021$ | $0.781 \pm 0.027$ | $0.190 \pm 0.015$ |
| KMeans (k = 1024) | $0.563 \pm 0.019$ | $0.938 \pm 0.007$ | $0.334 \pm 0.019$ |
| OTKE (k = 1024) | $0.878 \pm 0.015$ | $0.988 \pm 0.003$ | $0.838 \pm 0.066$ |
| Deep Sets (supervised) | $0.836 \pm 0.016$ | $0.981 \pm 0.002$ | $0.842 \pm 0.016$ |
| Set Transformer (supervised) | $0.884 \pm 0.019$ | $0.989 \pm 0.001$ | $0.851 \pm 0.015$ |
| MAESTRO (ours) | $\mathbf{0.923 \pm 0.014}$ | $\mathbf{0.992 \pm 0.003}$ | $\mathbf{0.897 \pm 0.029}$ |

Table 2: **Diagnosis Classification Benchmark.** MAESTRO outperforms all benchmarked methods across Accuracy, AUROC, and F1 metrics for diagnosis classification.

## D.2  SEX

| Model | Accuracy | AUROC | F1 |
|---|---|---|---|
| Manual Gating | $0.528 \pm 0.066$ | $0.540 \pm 0.074$ | $0.515 \pm 0.078$ |
| KMeans (k = 1024) | $0.631 \pm 0.029$ | $0.687 \pm 0.036$ | $0.601 \pm 0.052$ |
| OTKE (k = 1024) | $0.513 \pm 0.053$ | $0.521 \pm 0.072$ | $0.510 \pm 0.083$ |
| Deep Sets (supervised) | $0.682 \pm 0.058$ | $0.721 \pm 0.065$ | $0.665 \pm 0.071$ |
| Set Transformer (supervised) | $0.698 \pm 0.062$ | $0.756 \pm 0.059$ | $0.685 \pm 0.067$ |
| MAESTRO (ours) | $\mathbf{0.723 \pm 0.061}$ | $\mathbf{0.800 \pm 0.051}$ | $\mathbf{0.726 \pm 0.062}$ |

Table 3: **Sex Prediction Benchmark.** MAESTRO achieves superior performance across Accuracy, AUROC, and F1 metrics for sex prediction tasks compared to other methods.

## D.3  AGE

| Model | MAE | | $R^2$ | |
|---|---|---|---|---|
| Manual Gating | 19.96 | $\pm 39.98$ | $-0.670$ | $\pm 0.128$ |
| KMeans (k = 1024) | 18.74 | $\pm 34.58$ | $-0.370$ | $\pm 0.111$ |
| OTKE (k = 1024) | 8.46 | $\pm 18.69$ | $0.647$ | $\pm 0.099$ |
| Deep Sets (supervised) | 7.83 | $\pm 1.21$ | $0.662$ | $\pm 0.082$ |
| Set Transformer (supervised) | 7.46 | $\pm 1.06$ | $0.687$ | $\pm 0.054$ |
| MAESTRO (ours) | **6.70** | $\pm \mathbf{0.99}$ | **0.711** | $\pm \mathbf{0.088}$ |

Table 4: **Age Prediction Benchmark.** MAESTRO outperforms all benchmarked methods with the lowest Mean Absolute Error (MAE) and the highest $R^2$ score in age prediction tasks.

# E  APPENDIX FIGURES

## E.1  DATA

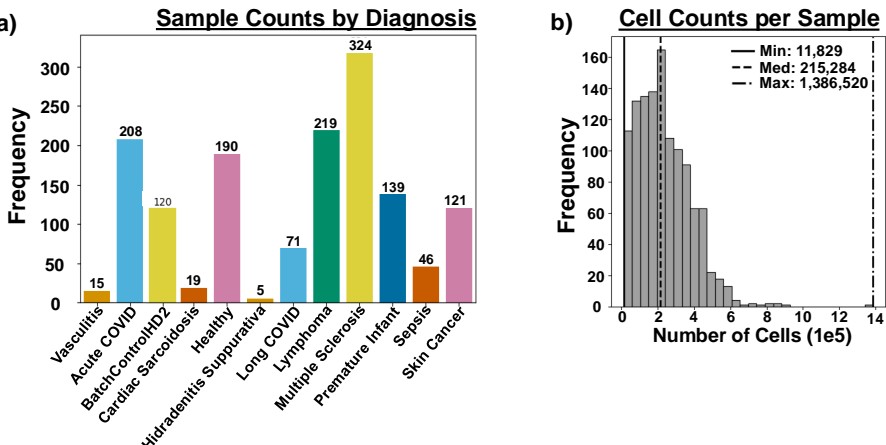

Figure 6: a) We show the counts of samples for each of the 11 different phenotypes (BatchControlHD2 is Healthy) used in our analysis. In total, there are 1,514 samples used. b) We show the distribution of number of cells per sample (the set size). The minimum set size is 11,829 cells and the maximum set size is 1,386,520 cells.

## E.2  DATA GENERATION: TIME AND PLACE

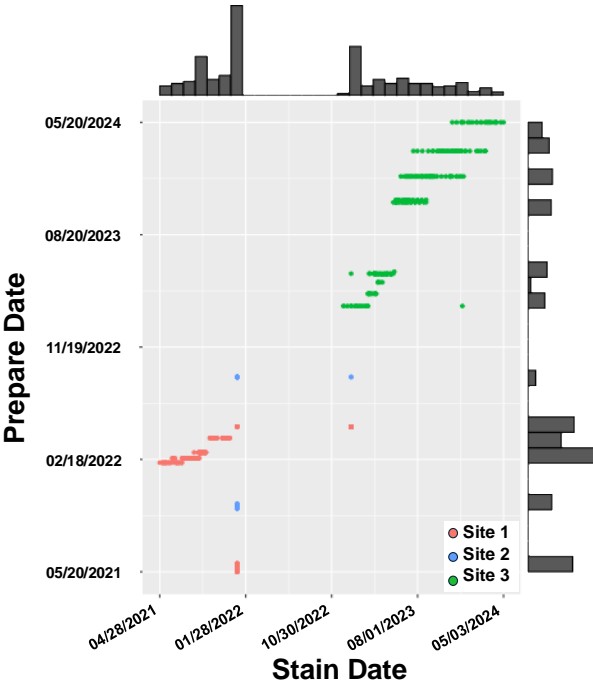

Figure 7: The x-axis denotes the date that the sample was stained with antibodies for protein marker detection, the y-axis denotes the preparation date, with histograms showing the distribution across these dates. The color denotes the location that the sample was processed which we keep anonymous for double blind review.

### E.3 BATCH EFFECTS

### E.3.1 ALL SAMPLES

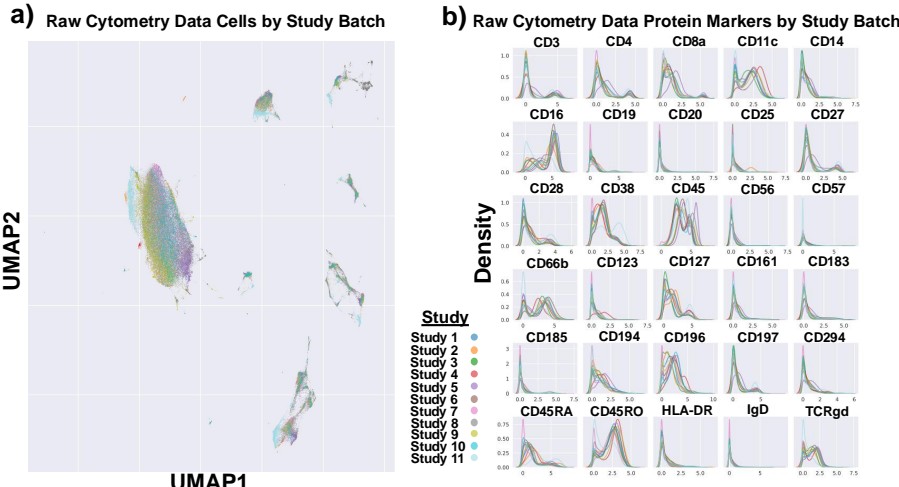

Figure 8: We randomly sample 1000 cells from all of our samples. a) We plot all of the cells on a UMAP colored by their study cohort. While the cells globally cluster by cell-type, we show that there are clear batch effects across the cohorts. b) We take the marker intensity distributions for each study and plot the density. This plot clearly demonstrates batch effects across the studies.

### E.3.2 BATCHCONTROLHD2 TECHNICAL CONTROL ONLY

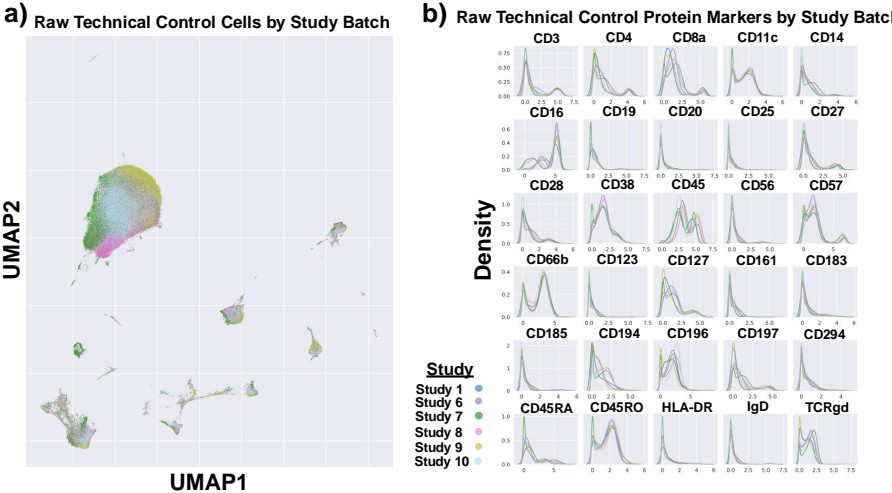

Figure 9: We use BatchControlHD2 as a technical control across the denoted studies. a) Here we randomly sample 1000 cells from each of these samples and plot them using on a UMAP which clearly show batch effects. b) We plot the distribution of protein marker expression for each study that BatchControlHD2 is in. Lastly, in Figure 3 we show that BatchControlHD2 clusters together demonstrating that there are less batch effects after processing with MAESTRO.

## F  APPENDIX INFORMATION

### F.1  DATA

We use CyTOF (Cytometry by Time of Flight) to generate our cytometry dataset. CyTOF captures high-dimensional data through mass cytometry, a technique that simultaneously measures up to 50 or more protein markers at the single-cell level. Unlike traditional flow cytometry, CyTOF uses metal isotopes conjugated to antibodies to bind specific cellular proteins. These metal-tagged antibodies are then detected using time-of-flight mass spectrometry, which minimizes signal overlap and allows for precise and comprehensive profiling of cellular phenotypes. This method facilitates in-depth analysis of complex biological systems, offering a detailed snapshot of cellular states and functions across large cell populations.

### F.2  PRE-TRAINING

Using 1,514 whole blood cytometry samples, we create a train and test set using an 80/20 random split within each diagnosis to ensure that all diagnoses are represented in the test set. This test dataset is never used for training during pre-training, linear probing, ablation, or cell-type proportion retrieval. We pre-train MAESTRO on four NVIDIA A100 GPU's for 156 hours at 29 minutes per epoch (324 epochs). Memory usage varies as each sample is of variable size, but on average the model consumes 58 GB per GPU. We train on our dataset with AdamW optimizer and a batch size of 1. Due to variable size inputs, we can only use a batch size of one (one for each GPU - four samples at once). The learning rate starts at 1e-4 and using a cosine annealing scheduler to a minimum of 1e-8. We use NRBM to mask four different copies of our samples at masking rates of 20%, 40%, 60%, and 80%. Masking is done on the fly during pre-training and not before-hand. We mask by replacing selected indices with a learnable mask token. MAESTRO is configured with four attention heads, a hidden size of 2,048, and a latent dimension of 1,024. Our input cells start at 30 dimensions (representing 30 different protein markers), which we use a linear layer to transform to 1,024 dimensions before using three ISAB (with 2,500 learned points per ISAB) blocks, followed by PMA. After pooling to a single learnable seed (vector), we decode the vector by copying the pooled input to each index of the unmasked cells and use the masking token to denote indices where the original cell was masked. We follow this with a PMA block with number of seeds equal to the size of the input matrix, and follow this with three SAB blocks. Finally we use a linear layer to transform our output to its original 30 dimensions. All attention blocks use a SwiGLU activation function to learn both linear and non-linear relationships (Shazeer, 2020). Reconstruction is calculated using Sinkhorn Optimal Transport with Euclidean distance for the cost matrix calculation. We align latent representations from the student and teacher model by using a non-linear projection head on the latent representation. Following the projection head we use softmax activation to convert our representations to probability distributions. We use Kullback-Leibler (KL) - Divergence to minimize the difference in distributions. The teacher model is updated with the exponential moving average of the student and a momentum value of 0.999. Student and teacher model temperatures are set to 0.1 and 0.07, respectively.

### F.3  BENCHMARKING METHOD IMPLEMENTATION

MAESTRO is pre-trained with an 80/20 train/test set. For linear probing, the same split is used for training and testing the logistic regression model in all benchmarked methods. The logistic regression model is implemented using scikit-learn LogisticRegression(). We repeat five-fold cross-validation 10 times for each method. Due to imbalances of diagnoses we report the Accuracy, AUROC, AUPRC, and F1-Score.

**MAESTRO**  MAESTRO is pre-trained according to the above. Following pre-training, latent representation for all samples are derived using the pre-trained MAESTRO model. Each representation is a vector with 1,024 dimensions. Following, the samples are split into the train and test groups as was used during pre-training to train and test a logistic regression model. The logistic regression model is fit on the training set for 14 various diagnoses, and then inference is done on the held-out test set.

**K-means**    We build two version of the K-means representation. One where we set k = 512, and the other k = 1024. We build these representations using the same $80\%$ of training samples used for all other methods. We perform k-means clustering on all available cells across the samples. Then, for each sample we calculate the proportions of each cluster that exist. This vector of cluster proportions is used as the input to a logistic regression model. The logistic regression model is fit on the training set for 14 various diagnoses, and then inference is done on the held-out test set. Due to imbalances of diagnoses we report the Accuracy, AUROC, AUPRC, and F1-Score.

**OTKE**    The OTKE method is unable to take variable sized sets and using large input sets overloaded the available GPU memory. For each sample, we use a random subset of 10,000 cells (equal to the number of cells input to the student model of MAESTRO). Then, we train the OTKE method, unsupervised, using our training samples. We use the default parameters as described by OTKE (Mialon et al., 2022). We then create representations for all of our samples using the learned kernel. We use the training samples to train a logistic regression model and report metrics on the held-out test set.

**Deep Sets**    As above, Deep Sets is subset to 10,000 cells (same as student model of MAESTRO). Deep Sets is inherently a supervised model so we directly train the model to predict the 14 diagnoses using the $80\%$ training set. We traing using hidden dimension of 2,048 (same as MAESTRO), and train until convergence or max 300 epochs (same as MAESTRO). We report metrics on the held-out test set.

**Set Transformer**    As above, Set Transformer is subset to 10,000 cells (same as student model of MAESTRO). Set Transformer is inherently a supervised model so we directly train the model to predict the 14 diagnoses using the $80\%$ training set. We traing using the same encoder architecture as MAESTRO (linear layer, three ISABs, and a PMA) using hidden dimension of 2,048 (same as MAESTRO), and a latent dimension of 1,024 (same as MAESTRO) and train until convergence or max 300 epochs (same as MAESTRO). We report metrics on the held-out test set.

### F.4    ABLATION IMPLEMENTATION

Unless otherwise denoted, all models in the ablation study use the same training parameters as the above section for MAESTRO pre-training. As above, we use the same 80/20 train/test split for our benchmarking studies.

**Random Masking**    We begin with the most simple model possible. We use only the masked autoencoding module of MAESTRO. We randomly subset each sample to 10,000 cells as we would with the Set Transformer. We randomly mask out elements of the set using a $50\%$ masking rate. We encode and decode each sample as described above in MAESTRO pre-training. After pre-training we derive representations for each sample using the pre-trained model. Using these representations, we use the $80\%$ training set from pre-training to also train a logistic regression model to predict our 14 diagnoses. We report metrics using the held-out test set.

**Multi-rate Masking**    We hypothesize that better representations can be learned using various masking rates, forcing the model to reconstruct missing cells using different amounts of information. Again, we randomly subset each sample to 10,000 cells as we would with the Set Transformer. We randomly mask out elements of the set at rates of $20\%$, $40\%$, $60\%$, and $80\%$. These values are arbitrarily chosen to cover the spectrum of $0\%$ to $100\%$. We encode and decode each sample as described above in MAESTRO pre-training. After pre-training we derive representations for each sample using the pre-trained model. Using these representations, we use the $80\%$ training set from pre-training to also train a logistic regression model to predict our 14 diagnoses. We report metrics using the held-out test set.

**Block Masking**    We hypothesize that our model is able to reconstruct the masked cells because they are randomly masked, and it's able to learn the reundancy of cells in an entire set. We implement NRBM to prevent our model from learning these relationships. Using the same implementation as the above multi-rate masking, we swap out random masking for NRBM. We encode and decode each sample as described above in MAESTRO pre-training. After pre-training we derive representations

for each sample using the pre-trained model. Using these representations, we use the $80\%$ training set from pre-training to also train a logistic regression model to predict our 14 diagnoses. We report metrics using the held-out test set.

**Self-Distillation** From the above we see minimal, although, improving performances for each added module. Here, we introduce our online tokenizer via self-distillation which allows for the encoding of the full set of cells. This is our full model and details on implementation can be found in the above section on MAESTRO pre-training.

**without Random Masking** Finally, to test that the self-distillation method itself is not what allows for rich representations in MAESTRO, we test performance using only the self-distillation module. We encode sets using a student teacher model and align the latents of the full set and subset of cells. Notably, there is a large decrease in performance when we remove the masked modelling section.

## F.5 CELL-TYPE PROPORTION RETRIEVAL

For each sample, 16 cell-type proportions were derived using manual gating, the gold-standard in immunology. We use these 16 cell-type proportions as the target for retrieval for each representation learning method. Since manual gating was used to derive these values, it is not benchmarked against in this section.

**MAESTRO** We pre-train MAESTRO using the above setting. After pre-training we derive each of the latent representations of each of our samples. Using the same 80/20 train/test split we train a one-layer neural network to simultaneously predict all 16 cell-type proportions. This is important because each value depends on the others. To demonstrate that this information is already encoded into the latent representation we only train for three epochs.

**K-means** K-means clustering is an alternative method to identifying cell-types, but is not considered the gold-standard like manual gating is. Because of this, k-means clustering serves as a strong baseline for other methods to outperform. Although we use k values of 512 and 1024 (well above the expected number of cell-types), we still expect this representation to be predictive of cell-type proportions. Using the same 80/20 train/test split we train a one-layer neural network to simultaneously predict all 16 cell-type proportions. This is important because each value depends on the others. To test whether information is already encoded into the latent representation we only train for three epochs.

**OTKE** We derive OTKE sample latent representations as described above. Using the same 80/20 train/test split we train a one-layer neural network to simultaneously predict all 16 cell-type proportions. This is important because each value depends on the others. To test whether this information is already encoded into the latent representation we only train for three epochs.

**Deep Sets** Deep Sets is supervised and therefore we do not use the one-layer neural network as described above. Instead, we use the default Deep Sets model with the 16 cell-type proportions as the target values. For each sample we subset their matrix randomly to 10,000 cells due to computational constraints. Using default parameters we train the model to predict 16 cell-type proportions. As above, because the information is inherently within the input we only use three epochs to train the model to test whether it is able to encode the 16 cell-types.

**Set Transformer** Set Transformer is supervised and therefore we do not use the one-layer neural network as described above. Instead, we use the Set Transformer encoder as described above, with the 16 cell-type proportions as the target values. For each sample we subset their matrix randomly to 10,000 cells due to computational constraints. Using default parameters we train the model to predict 16 cell-type proportions. As above, because the information is inherently within the input we only use three epochs to train the model to test whether it is able to encode the 16 cell-types.

## F.6 LIMITATIONS

MAESTRO faces several limitations that span computational, methodological, and interpretive challenges. The model has high computational demands, including long training times and reliance on advanced hardware such as multiple GPUs with high memory bandwidth, make it inaccessible to research groups with limited resources. MAESTRO requires single-sample batch processing due to variable input sizes limits optimization strategies, resulting in longer convergence times. Masking and reconstruction strategies like Non-Random Block Masking (NRBM) depend on specific assumptions about data redundancy, which may not generalize well to diverse datasets. The model's focus on global representations risks underrepresenting rare cell populations and outliers, which are biologically significant in many studies. Interpretability is another major concern, as the latent representations encode diagnostically relevant information but lack transparency in how specific features are prioritized, complicating biological insights. MAESTRO's dependency on fixed protein panels restricts its application to datasets with variable panels, while its computational intensity renders it unsuitable for real-time analysis. The absence of multi-modal data integration limits its adaptability to more complex biological studies involving diverse data types. Additionally, its reliance on manual gating for validation—a labor-intensive and operator-dependent process—hinders scalability for larger datasets. These limitations collectively underscore the need for further refinement of the model, increased validation across heterogeneous datasets, and improved accessibility and interpretability to expand its utility in biomedical research.

