# OpenReview forum: "MAESTRO: Masked Encoding Set Transformer with Self-Distillation"
_ICLR.cc/2025/Conference — ICLR 2025 Poster_

### Official Review · Reviewer_MoSb · 2024-10-31

**Soundness:** 3
**Presentation:** 3
**Contribution:** 4
**Rating:** 8
**Confidence:** 4

**Summary:**

MAESTRO introduces a self-supervised set transformer framework for analyzing cytometry data, which is known to be challenging due to its high dimensionality, permutation invariance, and variable sample sizes. Using masked encoding and a self-distillation approach, the model generates vector representations of immune profiles by leveraging attention mechanisms to handle set-structured data. MAESTRO performs better cell-type proportion retrieval and disease phenotype classification than state of the art techniques, improving single-cell analysis in immunology research.
Since cytometry datasets contain millions of cells per sample, MAESTRO proposes a three -fold strategy :
- A SSL set representation approach with a permutation-invariant attention mechanisms (using __ISAB, PMA, SAB__);
- A masked encoder (__NRBM__) that enables to process large cytometry datasets efficiently;
- A teacher-student model for self-distillation via __EMA__.

Thus, MAESTRO learns holistic representations of entire cell sets, capturing both global (sample-level) and local (cell-type) information.

**Strengths:**

- __Originality__: The paper presents a novel idea to deal with million-row size cytometry dataset, as single cells ones are, providing a model, MAESTRO, that can capture sample membership information without losing cell population-level interactions rather than uniquely focusing on individual cells.
- __Quality__: The manuscript clearly explains how to face the challenges of large single-cell cytometry datasets exploiting and combining previous-field ideas not deeply applied by compared models yet.
- __Clarity__: The paper is quite well written, with no major typos or incomprehension.
- __Significance__: The manuscript, presents a novel idea to face a common problem in a large cytometry dataset, and may well-impacts research due to its:
     - __SSL__ module that doesn’t require any label-acquisition process for training, which is particularly costly for these datasets;
     - ISAB, PMA, and SAB (__Eqs. 5-7__) that are finely tuned to maintain permutation invariance, a critical feature for handling unordered sets like these;
     - NRBM (__Alg.1__) rather than random masking to contain cell-populations level information.

**Weaknesses:**

- **Patient-batch limitations:**
   - The manuscript doesn’t address the problem of patient-normalization in scenarios where the model may have to deal with a heterogeneous cohort of patients. Cytometry patients' samples may vary a lot in a heterogenous cohort, and further studies on this generalization process could extend MAESTRO applicability (e.g. https://pubmed.ncbi.nlm.nih.gov/31633883/). For example, authors could specify whether would make sense to __inject__  __patient-level__ information as __prior knowledge__ during the pre-training phase.

-  **Scalability concerns with self-distillation on larger datasets and different batch sizes:**
   - This approach may become less effective as datasets start spanning over large patient cohorts since MAESTRO  has been pre-trained on four GPUs at a time, with corresponding  __batch_size=1__, meaning four samples at once have been processed. Under extremely large datasets, feeding the teacher model with complete sets can lead to substantial memory requirements.

-  **Dealing with noisy input:**
   - It’s not explicitly addressed the robustness of MAESTRO when dealing with noisy inputs, such as debris, dead cells that may be inherited from other cytometry datasets (e.g. flow cytometry ones), and whether this could or couldn’t be taken into account in the SSL strategy.

**Questions:**

Authors, in addition to the __above__ cited __perplexities__, may illustrate whether they have plans for exploring the following points:
   1. How does the __choice of protein markers__ affect MAESTRO’s performance and generalizability;
   2. How MAESTRO would perform on __multi-modal__ data types like epigenomic data, e.g. __ATAC-seq__;
   3. How MAESTRO’s embedding can support unsupervised tasks like __clustering__ or anomaly (__blast__ population) __detection__, in a potential __diagnosis__ scenario.

---

> ### Author Response · Authors · 2024-11-22
> **Official Rebuttal: Reviewer MoSb**
>
> Dear reviewer MoSb,
>
> Thank you for the kind, thorough, and constructive review of our paper. We address weakness 1 but otherwise assume that no changes need to be made to our manuscript. We will address the other weaknesses and questions that you have in your review here:
>
> **R4, Comment 1:**** Patient-batch limitations: The manuscript doesn’t address the problem of patient-normalization in scenarios where the model may have to deal with a heterogeneous cohort of patients. Cytometry patients' samples may vary a lot in a heterogenous cohort, and further studies on this generalization process could extend MAESTRO applicability (e.g. https://pubmed.ncbi.nlm.nih.gov/31633883/). For example, authors could specify whether would make sense to inject patient-level information as prior knowledge during the pre-training phase.
>
> **Response:** Great point! We’ve addressed this in the overall response to all authors but will reiterate here: we address this in lines 387-391, where we describe that our “single” dataset is actually a composition of many cohorts that were processed and generated at various time/location. We additionally provide supplemental figures E.1, E.2, and E.3 that demonstrate the batch effects apparent in the raw data. Lastly we use a technical control sample BatchControlHD2 which demonstrates batch effects in raw data, but are clustered together in Figure 3.
>
> **R4, Comment 2:** Scalability concerns with self-distillation on larger datasets and different batch sizes: This approach may become less effective as datasets start spanning over large patient cohorts since MAESTRO has been pre-trained on four GPUs at a time, with corresponding batch_size=1, meaning four samples at once have been processed. Under extremely large datasets, feeding the teacher model with complete sets can lead to substantial memory requirements.
>
> **Response:** This is a fair concern, however, because a feature of MAESTRO is that we can take variably sized inputs, there is little that can be addressed for batch sizes. We note that feeding the teacher model the complete set is not that memory intensive, and a novelty in our paper! The teacher model requires no gradient calculation as its parameters are the EMA of the student (which only receives the subset). Further, in our supplementary, we show the distribution of number of cells in our dataset, which on the upper end is over a million cells.
>
> **R4, Comment 3:** Dealing with noisy input: It’s not explicitly addressed the robustness of MAESTRO when dealing with noisy inputs, such as debris, dead cells that may be inherited from other cytometry datasets (e.g. flow cytometry ones), and whether this could or couldn’t be taken into account in the SSL strategy.
>
> **Response:** While we did not describe this in the paper, debris, dead cells, and doublets were all removed before-hand. While we believe that the removal of these are a pre-processing step, it is a great point to consider how/if we can bypass this step which would make the model even more impactful.
>
> **R4, Question 1:** How does the choice of protein markers affect MAESTRO’s performance and generalizability;
>
> **Response:** This is a great question, and we are unsure of the answer! It would be a great experiment to either test with a different and/or smaller protein panel, or, it could also be interested to test the result with masking of both cells (rows) as well as markers (columns)
>
> **R4, Question 2:** How MAESTRO would perform on multi-modal data types like epigenomic data, e.g. ATAC-seq;
>
> **Response:** We are excited and interested to test how MAESTRO performs on multi-modal data, but for this paper, is out of scope.
>
> **R4, Question 3:** How MAESTRO’s embedding can support unsupervised tasks like clustering or anomaly (blast population) detection, in a potential diagnosis scenario.
>
> **Response:** We believe that anomaly detection for populations such as blasts is better suited for models at the single-cell level, this is still a very interesting question. It might be interesting to plot the cells at an intermediate stage of the model (before pooling) to see how well these individual cells are represented. Since we demonstrated that our model embedding can do diagnosis/phenotype prediction, it’s intuitive to believe that anomaly populations are represented in a meaningful way.
>
> Thank you again for the great review of our paper! We assume we have addressed all concerns for paper that are within scope of this study. Thank you again for your time.

---

> > ### Comment · Reviewer_MoSb · 2024-11-25
> >
> > Thanks for uploading the rebuttal manuscript, and addressing my concerns as well. The final version, indeed, looks neat and well written. Thanks for addressing my more-general questions regarding MAESTRO applicability to broader scopes. I don’t have any more questions/concerns to be clarified. I therefore stick with my current score.

---

### Official Review · Reviewer_Jhp9 · 2024-11-02

**Soundness:** 2
**Presentation:** 3
**Contribution:** 2
**Rating:** 6
**Confidence:** 3

**Summary:**

In the paper, the authors proposed MAESTRO, a set-transformer-based method that design for generate the sample embeddings and cell embedding for the cytometry data. The authors compared several baseline method, conducted ablation studies, and further checked the effectiveness of the sample/cell embeddings on sample classification/cell type proportion retrieval.

**Strengths:**

The paper is generally well written and easy to follow.

The authors considered the specific needs for the cytometry data, such as variable set size, large data scale, and the permutation-invariant problem and designed their model, which is great and add a layer of novelty.

**Weaknesses:**

The downstream experiments presented in sections 4.3 and 4.4 of the manuscript, which focus on sample classification, are adequately performed. However, the section 4.5 dealing with cell type distribution retrieval does not meet the same standard. The rationale behind fine-tuning the embedding for this task is unclear, given the variability and sample-dependence of cell type distributions. This approach lacks the robustness required for generalization across different datasets.

Furthermore, the manuscript does not convincingly demonstrate the utility of the proposed embeddings in broader cytometry tasks. Downstream applications such as zero-shot cell classification, zero-shot sample characterization beyond disease/health state, and protein representation are notably absent. Incorporating these biologically meaningful experiments would significantly enhance the value and applicability of the research. More rigorous and diverse testing of the embeddings on a range of cytometry tasks is essential to establish their effectiveness and relevance in the field.

I did not get the biological importance on the permutation-invariant module. Any downstream tasks to show the effectiveness of the module?

**Questions:**

See weaknesses

---

> ### Author Response · Authors · 2024-11-22
> **Official Rebuttal: Reviewer Jhp9**
>
> Dear Reviewer Jhp9,
>
> Thank you for your thoughtful and detailed review of our manuscript. Below, we address your comments and provide clarifications where necessary:
>
> **R3, Comment 1:**
> The downstream experiments presented in sections 4.3 and 4.4 of the manuscript, which focus on sample classification, are adequately performed. However, section 4.5 dealing with cell-type distribution retrieval does not meet the same standard. The rationale behind fine-tuning the embedding for this task is unclear, given the variability and sample dependence of cell-type distributions. This approach lacks the robustness required for generalization across different datasets.
>
> **Response:** Thank you for raising this concern. The purpose of section 4.5 is to demonstrate that the information about cell-type distributions (which typically requires labeled data) is inherently stored within the embedding itself. We demonstrate this by using a neural network trained with a low number of epochs to predict the cell-type distribution. The use of fewer epochs highlights that the model already encodes this information and does not require extensive optimization.
>
> We acknowledge the inherent variability and sample dependence of cell-type distributions, and we show that our embeddings (which are similarly variable and sample-dependent) are capable of retrieving this information. Additionally, we understand that our language may have implied that we used a single dataset with minimal batch effects. As clarified in our general response to all reviewers, we address this on lines 387–391 by describing how our “single” dataset is, in fact, a composition of multiple cohorts processed at different times and locations. Supplemental figures (E.1, E.2, and E.3) further demonstrate batch effects in the raw data. Lastly, we utilize the technical control sample BatchControlHD2, which exhibits batch effects in the raw data but clusters together in Figure 3, demonstrating the robustness of our embeddings in the presence of batch effects.
>
> **R3, Comment 2:**
> Furthermore, the manuscript does not convincingly demonstrate the utility of the proposed embeddings in broader cytometry tasks. Downstream applications such as zero-shot cell classification, zero-shot sample characterization beyond disease/health state, and protein representation are notably absent. Incorporating these biologically meaningful experiments would significantly enhance the value and applicability of the research. More rigorous and diverse testing of the embeddings on a range of cytometry tasks is essential to establish their effectiveness and relevance in the field.
>
> **Response:** We appreciate your suggestion and have carefully considered the inclusion of such tasks. However, we believe that zero-shot cell classification is not aligned with the focus of this paper. The closest related task we address is cell-type distribution retrieval, which involves predicting an entire distribution of cell types from an embedding vector—a significantly more challenging task than single-cell classification.
>
> Regarding zero-shot sample characterization, we seek clarification on its definition in this context. Typically, zero-shot classification requires embeddings of classes obtained through external methods (e.g., embeddings from language models). If there are existing approaches for obtaining such embeddings for cytometry set data, we would be open to exploring them. However, we believe that this task falls outside the scope of our paper.
>
> As for protein representation, we also request clarification on its intended meaning in this context. Based on our understanding, this task does not pertain to the objectives or concepts explored in this paper.
>
> To address the concern of broader utility, we have provided additional evaluations beyond disease diagnosis, such as predicting sex and age, to showcase the discriminative power of our embeddings. These results are presented in Figure 4 and provide further evidence of their applicability.
>
> In summary, we have expanded our evaluations to reinforce the effectiveness of our embeddings and clarified our focus on set-level tasks rather than individual-cell-level analyses. We believe these updates and responses address your concerns. If there are any remaining issues or points of clarification, we would be happy to address them further.
>
> Thank you again for your time and constructive feedback!

---

### Official Review · Reviewer_72JE · 2024-11-02

**Soundness:** 3
**Presentation:** 3
**Contribution:** 2
**Rating:** 6
**Confidence:** 3

**Summary:**

The authors developed a method called MAESTRO (Masked Encoding Set Transformer with Self-Distillation) to effectively capture and summarize the diverse characteristics of immune cells from cytometry data. MAESTRO leverages a specialized attention mechanism and a self-distillation framework within a self-supervised learning setup, enabling it to handle large datasets without information loss. The model generates sample-level representations from the data. The authors evaluated MAESTRO’s embeddings to determine whether they can support downstream diagnostic classification, and enable cell-type proportion prediction.

**Strengths:**

The manuscript is well-written and easy to follow. The proposed method is effectively designed to handle large datasets without losing sample information. Additionally, the model addresses permutation invariance by using specialized attention blocks that omit positional encodings. This design enables MAESTRO to generate robust, representative embeddings for diagnostic classification and cell-type proportion prediction.

**Weaknesses:**

(1) The model was evaluated only on datasets from similar experimental settings, which contain minimal batch effects. It is unclear how the method handles batch effects or how the resulting embeddings may be influenced by such variations.
(2) According to the description, the detected proteins could be different between cells. Currently, the authors select and focus only on the shared detected proteins across all the samples. Could the model be extended to handle all the detected proteins?
(3) Additionally, the model primarily generates sample-level embeddings, whereas producing cell-level (for each cell) and feature-specific (for each feature) embeddings could be valuable for downstream comparisons.
(4) Further details on the method's runtime, robustness, and memory usage would also be beneficial.

**Questions:**

See the weakness section.

---

> ### Author Response · Authors · 2024-11-22
> **Official Rebuttal: Reviewer 72JE**
>
> Dear Reviewer 72JE,
>
> We sincerely thank you for your constructive and thoughtful review of our submitted paper. Below, we outline the modifications made to the manuscript and address additional points of clarification:
>
> **R2, Comment 1:**
> The model was evaluated only on datasets from similar experimental settings, which contain minimal batch effects. It is unclear how the method handles batch effects or how the resulting embeddings may be influenced by such variations.
>
> **Response:** This concern has been addressed in our general response to all reviewers. To reiterate, in lines 387–391, we clarify that our "single" dataset is, in fact, composed of multiple cohorts processed and generated at different times and locations. Supplemental figures (E.2, E.3.1, and E.3.2) illustrate the batch effects apparent in the raw data. Additionally, we highlight the use of our technical control sample, BatchControlHD2, which demonstrates batch effects in raw data but clusters together in Figure 3, showcasing how batch effects are handled in the latent space.
>
> **R2, Comment 2:**
> According to the description, the detected proteins could be different between cells. Currently, the authors select and focus only on the shared detected proteins across all the samples. Could the model be extended to handle all the detected proteins?
>
> **Response:** In this dataset, all samples processed independently contain the same 30 protein markers due to decisions made during the experimental process. Therefore, filtering for overlapping proteins was not necessary. While it is true that in a different experimental setup, samples might include non-overlapping protein markers, addressing such scenarios is outside the scope of this paper. However, we agree this is an interesting question and worth exploring in future work.
>
> **R2, Comment 3:**
> Additionally, the model primarily generates sample-level embeddings, whereas producing cell-level (for each cell) and feature-specific (for each feature) embeddings could be valuable for downstream comparisons.
>
> **Response:** We agree that cell-level and feature-specific embeddings are valuable; however, they are not the focus of this paper. The goal of our work is not to optimize the representation of individual cells but rather to generate the best representation of the collection of cells within a sample. We do not claim that individual cells are better represented by our method than by alternative single-cell models, such as scGPT. Unlike cell-level representation models, which require sample-level aggregation training, our approach directly generates sample-level embeddings without requiring this intermediate step. Furthermore, evaluations of single-cell models are typically based on validating cell type labels. In contrast, we demonstrate the ability to predict entire distributions of cell types (Section 4.5), which represents a more complex task than single-cell classification. We have clarified this distinction in lines 500–505, highlighted in red.
>
> **R2, Comment 4:**
> Further details on the method's runtime, robustness, and memory usage would also be beneficial.
>
> **Response:** Thank you for this suggestion. We have included details on runtime and memory consumption in Appendix F.2, highlighted in red, to provide a clearer understanding of these aspects.
>
> Please let us know if there are any remaining concerns or if additional revisions are required. Once again, we appreciate your valuable feedback and thoughtful review.

---

> > ### Comment · Reviewer_72JE · 2024-11-24
> >
> > Thank you for your response and clarification. I believe the authors solve all my concerns.

---

### Official Review · Reviewer_4TLg · 2024-11-03

**Soundness:** 3
**Presentation:** 3
**Contribution:** 2
**Rating:** 6
**Confidence:** 3

**Summary:**

This paper presents MAESTRO, a self-supervised learning method tailored to learn representations of high-throughput cytometry data. The complexity and variability of the data makes it impossible to directly apply many of the previously developed techniques, so the authors come up with a new method using the existing teacher-student architecture to learn representations of immune profiles. The authors present evidence of effective data reconstruction, probe representations in predicting sample diagnosis and cell type proportion, and demonstrate superior performance to existing techniques. In addition, the authors report results of the ablation study to justify the design of MAESTRO.

**Strengths:**

- I appreciated the detailed and comprehensive method description. Clearly structured and explained in sufficient detail. Although many of the blocks are not novel, such a presentation helps understanding the work.
- The contribution of the work looks solid, as demonstrated in the experiments. The design of the proposed method is justified with an ablation study. The performance appears superior to previously proposed methods.
- Arguably, there exist few solutions capable of handling high-throughput cytometry data to this day. MAESTRO seems to make a significant contribution in the domain by tackling this challenge effectively.

**Weaknesses:**

1. It seems that the work addresses an existing task and tackles it by integrating existing concepts and approaches into a new framework. Therefore, the novelty of this work appears limited and must be further clarified by the authors.
2. Evaluation is done on a single dataset, which is generally not enough to showcase the effectiveness and robustness of the newly presented method. The cited DeepCyTOF, for example, employed five collections of FCM datasets from FlowCAP-I and three additional collections of CyTOF datasets.
3. Data and code availability are not discussed. For a method paper, an anonymized repository must be provided for reviewers to verify the soundness and validity of the approach.
4. The authors cite the paper of [cyMAE](https://www.biorxiv.org/content/10.1101/2024.02.13.580114v2) to claim that manual gating remains state of the art, while this very method was introduced at the NeurIPS 2023 Workshop AI4Science as the first effort to achieve (and, arguably, surpass) this state-of-the-art performance. Comparison to cyMAE is neither presented, nor discussed, which is a questionable choice of the study design.
5. Only a few concluding remarks are dedicated to the limitations of the approach. More discussion points could follow from the additional evaluations that are currently missing.
6. References look limited suggesting the authors might not be aware of the other important works in the field. Also, some statements are missing citations (e.g., lines 83-92), which complicates validity assessment.
7. Minor flaws:
- line 296: missing bracket typo
- line 313: double-quote typo
- line 317: “Algorithm 0” typo

**Questions:**

__Contributions__

1. Is MAESTRO tailored to the analysis of immune profiles? How well can it generalize beyond that? What would be the evidence of that? If there are no additional experimental results, please discuss potential applications of MAESTRO to set-structured data outside of immunology, and what modifications, if any, might be needed for such applications.
2. What is the strongest argument to defend the novelty of this work?

__Figure 3b__

3. How do you explain values for Sepsis, Vasculitis, and two types of COVID?

__Table 1__

4. If manual gating performs so poorly, why is it called the golden standard? Please discuss reasons it remains widely used despite the emergence of more accurate computational methods and consider abstaining from calling it a gold standard.
5. The table includes 2 methods that are supervised. However, the [cyMAE paper](https://www.biorxiv.org/content/10.1101/2024.02.13.580114v2)  suggests that it is gradient boosting decision trees (GBDT) that achieve top performance among the supervised learning algorithms. Why is there no comparison to GBDT?
6. Is a single linear probing task enough to evaluate the discriminative power of the learned representations? Is it possible they are biased towards sample diagnosis? Please include other evaluation tasks to provide a more comprehensive assessment of the learned representations.

__Evaluations__

7. Some other methods have been included for comparison despite the fact that they are incapable of handling large datasets. To make it possible, the authors sampled 10k cells for each sample ranging from 11k to 1386k cells in total. How fair and informative is that comparison? Were the other methods optimized to achieve their top performance under such conditions? Is it possible to compare MAESTRO to the other methods on a subset of the large dataset under entirely identical conditions? Please provide a more detailed justification for the comparison methodology.

---

> ### Author Response · Authors · 2024-11-22
> **Official Rebuttal: Reviewer 4TLg (1/3)**
>
> Dear Reviewer 4TLg,
>
> We sincerely thank you for taking the time to provide a fair and constructive review of our work. Below, we outline how we have addressed each of your comments:
>
> **R1, Comment 1:**
> It seems that the work addresses an existing task and tackles it by integrating existing concepts and approaches into a new framework. Therefore, the novelty of this work appears limited and must be further clarified by the authors.
>
> **Response:** Our paper was submitted under the Primary Area of “Applications to physical sciences (physics, chemistry, biology, etc.).” We respectfully argue that this area of submission inherently broadens the criteria for novelty. However, we understand that ICLR focuses primarily on methodological contributions. To address this, we emphasize that our work explores an uncharted task—set-level representation learning at the scale of millions of rows—which is an entirely new frontier. While our framework integrates existing concepts, we believe that incremental innovations often lay the foundation for impactful advancements in science. Furthermore, there are currently no existing models capable of solving the task our model addresses, which underscores its novelty.
>
> **R1, Comment 2:**
> Evaluation is done on a single dataset, which is generally not enough to showcase the effectiveness and robustness of the newly presented method. The cited DeepCyTOF, for example, employed five collections of FCM datasets from FlowCAP-I and three additional collections of CyTOF datasets.
>
> **Response:** As addressed in our general response above, we clarify in lines 387–391 that our “single” dataset is, in fact, composed of multiple cohorts, with samples processed and generated at different times and locations. To further address this concern, we provide supplemental figures (E.2, E.3.1, and E.3.2) that illustrate batch effects in the raw data. Additionally, we highlight the use of our technical control sample, BatchControlHD2, which demonstrates batch effects in raw data but clusters together in Figure 3.
>
> **R1, Comment 3:**
> Data and code availability are not discussed. For a method paper, an anonymized repository must be provided for reviewers to verify the soundness and validity of the approach.
>
> **Response:** We have made our codebase anonymously available for reviewers. The link to the repository is included in the abstract. Thank you for this suggestion!
>
> **R1, Comment 4:**
> The authors cite the paper of cyMAE to claim that manual gating remains state of the art, while this very method was introduced at the NeurIPS 2023 Workshop AI4Science as the first effort to achieve (and, arguably, surpass) this state-of-the-art performance. Comparison to cyMAE is neither presented nor discussed, which is a questionable choice of the study design.
>
> **Response:** We have updated the citation of cyMAE to its most recent publication in Cell Reports Medicine. Additionally, we have included further citations to support the use of gating as ground-truth labels for individual cells. We have adjusted the language to clarify that while cyMAE operates at the single-cell level, it validates the use of manual gating as state-of-the-art, given that it uses the same data type. We have not benchmarked against cyMAE because a comparison to our set-level model would be inherently unfair. Please refer to lines 130–134 (highlighted in red) for these revisions.
>
> **R1, Comment 5:**
> Only a few concluding remarks are dedicated to the limitations of the approach. More discussion points could follow from the additional evaluations that are currently missing.
>
> **Response:** In addition to addressing your evaluation comment by providing new evaluations of sex and age, we have included an extended limitations section in the supplementary material (F.6, lines 1406–1422).
>
> **R1, Comment 6:**
> References look limited, suggesting the authors might not be aware of other important works in the field. Also, some statements are missing citations (e.g., lines 83–92), which complicates validity assessment.
>
> **Response:** Thank you for identifying this issue. We have added citations to lines 83–92 and have expanded the reference list to ensure our work can be assessed rigorously for validity.
>
> **R1, Comment 7:**
> Minor flaws:
> Line 296: missing bracket typo
> Line 313: double-quote typo
> Line 317: “Algorithm 0” typo
>
> **Response:** Thank you for pointing out these typographical errors. We have corrected them in the revised manuscript.

---

> ### Author Response · Authors · 2024-11-22
> **Official Rebuttal: Reviewer 4TLg (2/3)**
>
> **R1, Question 1:**
> Is MAESTRO tailored to the analysis of immune profiles? How well can it generalize beyond that? What would be the evidence of that? If there are no additional experimental results, please discuss potential applications of MAESTRO to set-structured data outside of immunology, and what modifications, if any, might be needed for such applications.
>
> **Response:** Thank you for raising this question. While we believe that MAESTRO has the potential to generalize beyond immune profiles, we have not extensively tested this aspect yet. We feel that demonstrating this is outside the scope of the current paper, particularly given the page limitations, but we acknowledge its importance. In the conclusion section, we suggest several potential applications of MAESTRO to set-structured data outside of immunology, such as scRNA-seq, computational histopathology, and multi-modal integration. We have addressed this to the best extent possible without sacrificing critical components of the paper.
>
> **R1, Question 2:**
> What is the strongest argument to defend the novelty of this work?
>
> **Response:** The strongest argument for the novelty of this work is that no existing methods can represent sets of millions of rows with variable sizes as fixed vectors. Additionally, no current methods produce patient-level (as opposed to single-cell-level) representations. Our use of self-distillation to address the scale of cytometry data is a novel and validated approach to encoding million-cell-sized samples. Further, attention-based self-supervised set representation models are not yet present in the literature, making MAESTRO a unique contribution. Lastly, cytometry provides a different perspective of the patient than scRNAseq does, as it represents protein markers (after post translational modification), and therefore is an extremely relevant modality of data clinically, in which we show we have the ideal patient representation for single-cell set structured data.
>
> **R1, Question 3:**
> How do you explain values for Sepsis, Vasculitis, and two types of COVID?
>
> **Response:** This result suggests that certain pairs of diseases share underlying biological mechanisms, which manifest in similar patterns off the diagonal in the figure. For example, sepsis and acute COVID are both acute, life-threatening conditions characterized by significant immune dysregulation, such as T cell activation.
>
> **R1, Question 4:**
> If manual gating performs so poorly, why is it called the gold standard? Please discuss reasons it remains widely used despite the emergence of more accurate computational methods and consider abstaining from calling it a gold standard.
>
> **Response:** Manual gating is considered the gold standard because, despite its limitations, there are no other widely accepted methods for this task. While set-learning methods such as DeepSets, Set Transformer, and OTKE exist, our implementation is the first to apply them to cytometry data. Manual gating relies on biological priors, as opposed to unsupervised or data-driven approaches. Additionally, factors such as panel choice and sample processing conditions introduce technical variability, which unsupervised methods may misinterpret as noise patterns. Immunologists account for this variability by manually adjusting gates. While manual gating is the gold standard for vector representations of cytometry data, in the context of unsupervised, data-driven methods, the standard would be calculating proportions of clusters (as demonstrated in our paper using k-means). Lastly, changes in cell type proportion (determined through gating) is a standard signature/biomarker to use for understanding of immune status change in immunology. Standard analysis in immunology is using manual gating to obtain cell type proportion and use it for case-control analysis. We have revised the manuscript (lines 131–134, highlighted in red) to provide a more nuanced discussion.
>
> **R1, Question 5:**
> The table includes two methods that are supervised. However, the cyMAE paper suggests that gradient boosting decision trees (GBDT) achieve top performance among supervised learning algorithms. Why is there no comparison to GBDT?
>
> **Response:** The cyMAE paper and its benchmark of GBDT are conducted at the single-cell level, whereas our work operates at the set level. As such, these methods are not directly comparable to MAESTRO. The supervised methods included in our comparisons are specifically designed to work at the set level, aligning with the scope of our study.

---

> ### Author Response · Authors · 2024-11-22
> **Official Rebuttal: Reviewer 4TLg (3/3)**
>
> **R1, Question 6:**
> Is a single linear probing task enough to evaluate the discriminative power of the learned representations? Is it possible they are biased towards sample diagnosis? Please include other evaluation tasks to provide a more comprehensive assessment of the learned representations.
>
> **Response:** We have addressed this in our general response to all reviewers. Specifically, Figure 4 has been updated to include additional evaluation tasks. Additionally, we emphasize that the cell-type distribution prediction task in our paper is highly relevant. This task predicts the product of manual gating, which performs poorly in diagnosis prediction, demonstrating that the learned representations are not biased toward diagnosis.
>
> **R1, Question 7:**
> Some other methods have been included for comparison despite being incapable of handling large datasets. To make it possible, the authors sampled 10k cells for each sample ranging from 11k to 1,386k cells in total. How fair and informative is that comparison? Were the other methods optimized to achieve their top performance under such conditions? Is it possible to compare MAESTRO to the other methods on a subset of the large dataset under entirely identical conditions? Please provide a more detailed justification for the comparison methodology.
>
> **Response:** We recognize that this comparison may seem unfair; however, it highlights one of MAESTRO's key novelties—its ability to handle datasets far larger than 10k cells. Existing methods like Set Transformer experience memory issues with datasets exceeding 10k cells, making them unsuitable for large-scale applications. While it is possible to downsample MAESTRO to 10k cells for a direct comparison, one of the defining strengths of our method is its capacity to handle the full dataset, which we believe makes such a comparison unjustified.
>
> Once again, we deeply appreciate your thoughtful feedback and believe we have addressed all your concerns in the revised manuscript. If there are any remaining issues, we would be happy to make further revisions. Thank you for your time and valuable comments!

---

> > ### Comment · Reviewer_4TLg · 2024-11-24
> >
> > I sincerely thank the authors for making efforts to clarify and resolve my concerns!
> >
> > Being an applied AI scientist and thanks to the received comments, as well as valuable additions to the manuscript, I intend to raise my score to the above the acceptance threshold. Despite the limitations and the nuanced evaluation, I find the manuscript solid and the contribution good enough.

---

### Author Response · Authors · 2024-11-22
**Rebuttal Overall**

We would like to sincerely thank the reviewers for their time and insightful feedback. We are encouraged by the positive reception of our work and are eager to address the remaining points of concern. Overall, we have made every effort to address all of the issues raised by the four reviewers, and we provide detailed responses to each reviewer individually, outlining where and how we have addressed their comments. All changes made in response to reviewer feedback are highlighted in red. Below, we summarize the most commonly raised points for clarity:

***Batch effects:***
We have revised the language in the data section to more accurately and comprehensively describe the acquisition of our data. While we initially referred to it as a "single" dataset, it is more accurately described as a dataset composed of multiple cohorts of patients, with samples processed and generated at different locations and times. To address this, we have included supplementary figures that illustrate the batch effects in our data (Appendix E.2, E.3.1, E.3.2). Additionally, we specifically highlight the use of our technical control sample, BatchControlHD2, which is utilized across various cohorts. This sample demonstrates batch effects but clusters together in the latent space. The revised language can be found on lines 387–391 and 444–448.

***Insufficient probe evaluation:***
We have expanded the evaluation of our model to include tasks beyond disease diagnosis. In particular, we show that MAESTRO outperforms benchmarked methods in predicting sex and age, which are critical components of an individual’s immune status. We have replaced Table 1 with Figure 4 (as well as supplementary Table 2,3,4) to present the performance results more effectively, as the table could not accommodate the additional data. New text addressing these updates can be found in Section 4.4 (lines 455–470).

Furthermore, we have clarified the importance of the cell-type distribution prediction task. This task is of critical relevance for the following reasons:

1. Cell-type distribution is the output of manual gating, and demonstrating that this information is preserved in our latent embedding highlights the utility of our model.
2. Our model is the first self-supervised learning (SSL) cytometry model operating at the set level, tackling the significantly more complex task of predicting entire distributions compared to other SSL models, which focus on predicting single-cell types. The revised language for this section is located on lines 500–505.
3. Since it is a representation of a set of cells, it should theoretically carry information about the proportions of cell types. This experiment tests whether our representation is a good representation of the immune status of a sample as well as a set of bunch of cells.

We will provide additional detailed responses to individual reviewer comments below. Once again, we deeply appreciate the reviewers’ thoughtful feedback and are happy to address any further concerns.

---

### Author Response · Authors · 2024-12-03
**Authors final remarks and rebuttal period summary**

We sincerely thank the reviewers for their participation and insightful comments, which have greatly enhanced our paper. We believe our paper makes contributions that are beneficial to the machine learning community as a whole as well as the biomedical community within. Below is a summary of the rebuttal and discussion for the Primary and Area Chairs:

We are encouraged by the positive scores of 8, 6, 6, 6 with corresponding confidence scores of 4, 3, 3, 3.

Reviewers highlighted several key strengths, including the originality of our novel architecture, which uniquely handles extremely large, permutation-invariant sets—capabilities unmatched by existing models. Multiple reviewers praised the manuscript's quality, noting it is well-written and easy to follow. The significance of our work was recognized for addressing a previously unexplored problem and introducing a new deep learning architecture applicable to other large set challenges. These strengths are further emphasized by our solid contributions, demonstrated through sound experiments and improved performance over previous methods.

Key concerns during rebuttal were batch effect mitigation (generalizability) and evaluation sufficiency. In the revised paper (changes highlighted in red and blue), we clarify that our dataset includes multiple cohorts, demonstrate the presence of batch effects in supplementary figures, and show through technical controls that representations are similar across batches. We also add evaluations for age and sex, showcasing strong immune system representation, and refine our language to highlight our model's utility and importance.

We are pleased with the rebuttal outcomes and deeply appreciate the reviewers' time. We also thank the PCs, SACs, and ACs for their dedication and contributions to the community.

---

### Meta-Review · Area_Chair_BrjU · 2024-12-31

**Metareview:**

The authors developed a new tool for analyzing biological cytometry data, MAESTRO, a self-supervised model for immune cell profiling. The tool operates on sets level to generate sample-level representations. Using attention mechanisms and a self-distillation tokenizer, it outperforms existing methods in retrieving cell-type annotations across samples (including samples with batch effects and noises), and identifying clinically relevant features for disease diagnosis and immune state characterization.

- On positive side, this work is a strong, well organized/presented work that apply existing approach to a key biological data type, with reasonable ablation and comparative studies on a large dataset, which will facilitate the subject area research (immune cell profiling) via providing a useful method.
- On negative side, nonetheless, the reviewers highlighted that the underlying method is mainly based on existing approach and ideas (e.g. Mialon et al, cited as OTKE had very competitive performance, with attention-like mechanism, and was showcased for a broader range of tasks. Also, this work is limited in its scope to a fairly narrow subject, and the authors were not able to show broader applicability of the method beyond immune cell context, which should be improved further when they are releasing the code/method in its final format.

Overall given that this data type is of key clinical relevance and the method help to provide a useful tool, the work is considered generally to be weakly acceptable.

**Additional Comments On Reviewer Discussion:**

The discussions were robust in identifying key strengths and weaknesses, noting the importance of having an approach operating at sample-level with reasonable scalability, and presented as a domain-specific method for clinical relevant analysis. Yet the reviewers also unanimously raised the concern that there is limited novelty of the foundation method, and the limited testing done was a weakness that may leave room to desire for a broader audience. As the authors were able to highlight the utility and motivation, and added clarity on the datasets and comparisons, the reviewers were overall positive after the revision.

---

### Decision · Program_Chairs · 2025-01-22

Accept (Poster)